# The presence of ancient subducted oceanic crust contributes to seismic anomalies in Large Low Shear Velocity Provinces

Ewa Krymarys ✉, Motohiko Murakami ✉, Pinku Saha & Christian Liebske

Large Low Shear Velocity Provinces (LLSVPs) near the core-mantle boundary (CMB) are key yet enigmatic structures. Their origin is often linked to the accumulation of subducted mid-ocean ridge basalt (MORB), but computational models question MORB as the sole source due to its predicted high shear wave velocity compared to normal mantle. This uncertainty is compounded by the lack of direct sound velocity measurements at CMB pressures. Here we address this gap through ultrahigh-pressure shear wave velocity measurements on $CaCl_2$- and $\alpha$-$PbO_2$-type $SiO_2$, major phases in MORB, at pressures exceeding those of the CMB. Our results show shear velocities in dense $SiO_2$ phases are ~ 7–14% lower than previous predictions under these conditions. Incorporating these values into MORB models suggests that the typical seismic anomaly of $-1.5\%$ ($\delta \ln V_S$) observed in LLSVPs can be explained by ~ 23–33 vol.% oceanic crust along a cold slab geotherm, without invoking extreme thermal anomalies (+1500 K). Considering a subduction history exceeding 2 billion years, this scenario supports long-term MORB accumulation at the lowermost mantle. These findings provide new constraints on LLSVP composition and offer critical insights into deep mantle dynamics and the evolution of Earth's interior.

## Main Text

The Large Low Shear Velocity Provinces (LLSVPs) are recognized as the most extensive seismic anomalies in the lower mantle, exhibiting negative shear wave velocity anomalies ($\delta \ln V_S = \delta V_S/V_S * 100\%$) of approximately 0.5% to 3.0% relative to the surrounding averaged seismic velocities[1–9]. These seismic anomalies are observed beneath Africa and the Pacific and extend thousands of kilometers horizontally and hundreds of kilometers vertically above the CMB region[10,11]. Clarifying the origin of the LLSVPs, which give rise to one of the largest seismic heterogeneities in the Earth's mantle, is thus considered essential for gaining insights into the dynamics and evolution of Earth's interior throughout its history. A number of possible origins for LLSVPs have been proposed to satisfy physical and chemical properties as inferred from seismic features, that are manifested as thermal, chemical, or thermochemical variations[5,12]. Waveform- and travel-time-based seismological studies that indicate a sharp decrease in shear wave velocity at the edges of LLSVPs[12–14] suggest that the seismic characteristics of the LLSVPs

are more plausibly explained by the presence of chemically distinct materials compared to the surrounding mantle[3,15–17], rather than solely being caused by a high-temperature anomaly[12]. However, the possibility of a combination of thermal and chemical factors still cannot be fully ruled out.

### Hypothesis: Accumulation of ancient subducted oceanic crust in the lowermost mantle

One of the most widely supported hypotheses among the models proposing that LLSVPs are composed of chemically distinct entities suggests that their origin is attributed to the long-term accumulation of ancient subducted oceanic crust in the lowermost mantle[3,18,19]. This hypothesis is supported by geodynamical simulations that predict accumulation of subducted oceanic crust to accumulate in LLSVPs regions[2,8,20]. The density of oceanic crust determined from high-temperature and high-pressure experiments was shown to be large enough to ensure gravitational stability in the lowermost mantle[3,8,18]. This implies that LLSVPs inferred from the accumulation of

Department of Earth and Planetary Sciences, Institute of Geochemistry and Petrology, ETH Zürich, Zürich, Switzerland.
✉e-mail: ewa.krymarys@eaps.ethz.ch; motohiko.murakami@eaps.ethz.ch

subducted oceanic crust could potentially maintain seismic heterogeneity persistently.

The first crucial step in verifying the plausibility of this hypothesis is to directly compare seismic wave velocities of subducted oceanic crust in the lowermost mantle with the seismic structures of the LLSVPs. However, in-situ high-pressure velocity measurements of constituent minerals of mid-ocean ridge basalt (MORB), which are bridgmanite (Bd), $SiO_2$, $CaSiO_3$-perovskite (CaPv), and Ca-ferrite-type phase (CF)[18], at the extreme conditions down to the lowermost mantle, are rather limited. This hypothesis has thus so far primarily relied on verification through computational calculations[21–25] or mineral physics approaches based on the elastic properties determined under ambient or very low-pressure conditions[26–29]. Both mineral physics approach and recent ab initio calculations[30,31] have demonstrated that the shear wave velocity of MORB becomes higher than that of the surrounding mantle, under lowermost mantle conditions. This implies that the subducted oceanic crust alone might not be a plausible source for LLSVPs. Therefore, it has been suggested that explaining LLSVPs through MORB incorporation would require unrealistically high-temperature anomalies of 1500–1600 K at most[8,32]. On the other hand, recent measurements of elastic wave velocities for cubic CaPv[26,27], conducted up to ~15–20 GPa, have indicated that the shear wave velocity of CaPv is slower than previously predicted from the computational works. This observation raises the compelling possibility that the reduced shear wave velocity of MORB induced by the CaPv could explain LLSVPs. Nevertheless, the determined stability field of CaPv, starting above 20 GPa[33–35] remains experimentally unverified. Following the most recent first-principles calculations of[36], the stability field of cubic CaPv starts at 50 GPa, 1200 K and remains stable throughout lower-mantle conditions. Furthermore, the substantial discrepancies between theoretical data[37,38] and experimental results obtained under similar experimental setups and pressure conditions[26,27,39] undermine the reliability of a robust argument on the seismic structures of the LLSVPs.

Among the constituent minerals of MORB, the $SiO_2$ phase is thought to be the fourth most abundant major mineral, comprising ~17–20 vol.% in MORB assemblages[18,40,41]. However, within primary minerals present in the lower mantle, the $SiO_2$ phase has been believed to be the hardest (in bulk and shear moduli) mineral, far surpassing even bridgmanite[42]. In addition to this, due to reported slow silicon diffusion[43], it can preserve its hardness even at high temperatures, which implies that its shear wave velocity is considered one of the fastest among minerals in the Earth's mantle. Therefore, in order to assess the shear wave velocity nature of MORB under lowermost mantle conditions, obtaining shear wave velocity data for the $SiO_2$ phase down to the CMB is arguably a crucial experimental challenge that needs to be addressed. However, elastic wave velocity measurements for the high-pressure polycrystalline $SiO_2$ phase has only been determined up to ~70 GPa and never been explored down to the lowermost mantle pressures[28,44]. A predominant high-pressure $SiO_2$ phase, which is believed to have a $CaCl_2$-type structure, is stably present under most of the lower mantle conditions. This phase is known to further undergo a transition to an α-$PbO_2$-type structure under lowermost mantle conditions above ~120 GPa and 2400 K[45–49]. According to previous computational studies[22–24,50], the shear wave velocity of the $CaCl_2$-type $SiO_2$ phase is notably higher than that of other mantle mineral phases. This suggests that it significantly contributes to the overall increased shear wave velocity structure of MORB, compared to the surrounding mantle. On the other hand, there is also a computational prediction suggesting that the transition from the $CaCl_2$-type phase to the α-$PbO_2$-type phase could cause a sharp drop in shear wave velocity of approximately 1–2% at the transition pressure[24]. This could possibly lead to a reduction in MORB's shear velocity at the very bottom of the lower mantle. Yet, this phenomenon has not been experimentally verified. Therefore, making a quantitative assessment whether the ancient subducted MORB, which sank to the lowermost mantle, can indeed explain the seismic characteristics of LLSVPs, remains challenging.

## In-situ ultrahigh-pressure shear wave velocity measurements of $SiO_2$ phases

To address this issue, we determined the shear wave velocities of both the $CaCl_2$-type and α-$PbO_2$-type $SiO_2$ phases under pressure conditions up to

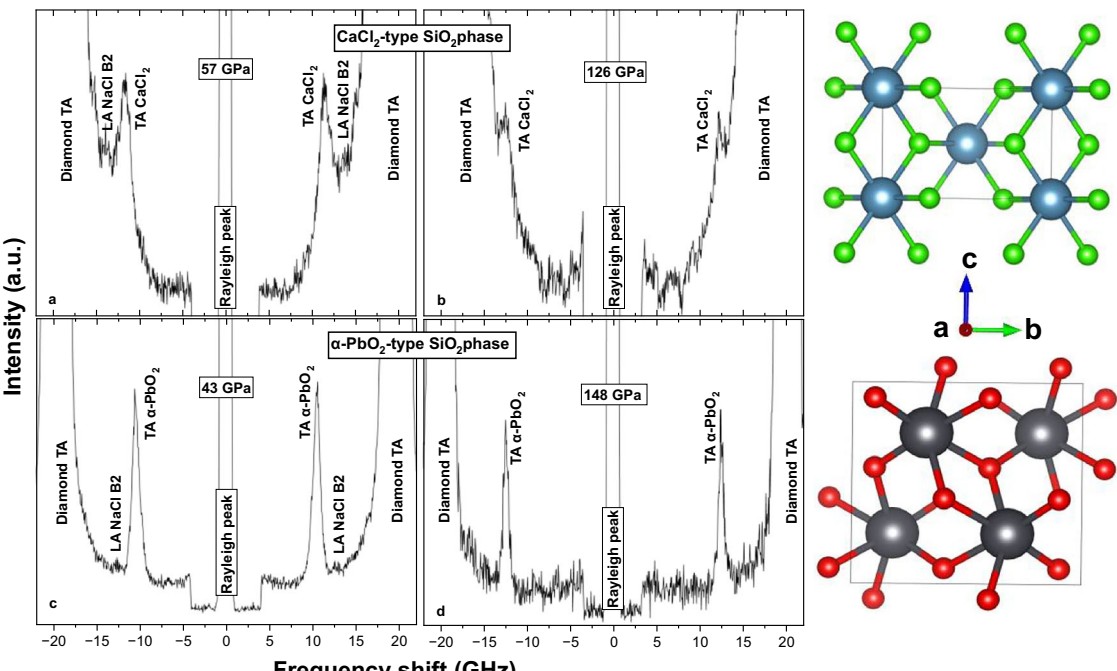

**Fig. 1 | Brillouin spectra of high-pressure $SiO_2$ phases loaded with NaCl B2 as pressure medium in the lower mantle. a, b** Polycrystalline $CaCl_2$-type $SiO_2$ phase at 57 and 126 GPa and 300 K. **c, d** Polycrystalline α-$PbO_2$-type $SiO_2$ phase at 43 and 148 GPa and 300 K. Atomic distributions in the orthorhombic lattices of $CaCl_2$-type (upper, green) and α-$PbO_2$-type (lower, red) structures, where dark green/grey atoms represent silicon and green/red atoms represent oxygen[89].

148 GPa, well surpassing those of the lowermost mantle, using in-situ ultrahigh-pressure Brillouin scattering technique with a diamond anvil cell (DAC). This was conducted in combination with synchrotron X-ray diffraction (XRD) for high-pressure structural and stress analysis, as well as Raman scattering measurements (see Fig. 1, Materials and Methods, Figure S1–S5, Table S1). Two different starting materials were used to synthesize the $CaCl_2$-type and $\alpha$-$PbO_2$-type $SiO_2$ phases, respectively. The samples were employed to investigate the elastic properties of the high-pressure phases of $SiO_2$ in the pure system and to discuss its implications in a MORB composition. The $CaCl_2$-type $SiO_2$ phase was synthesized in a DAC by compressing stishovite – pre-synthesized in a large-volume press – above 55 GPa. The $\alpha$-$PbO_2$-type $SiO_2$ phase was directly synthesized by compressing synthetic $\alpha$-cristobalite starting material above around 40 GPa in a DAC, following previous experimental studies[47,51–54]. The high-pressure $SiO_2$ phases were confirmed as the intended target high-pressure phases through in-situ high-pressure synchrotron X-ray diffraction measurements (Figure S3, S4). The obtained lattice constants and unit cell volumes were found to be in good agreement with those reported by Grocholski et al.[48] and supplemented with Raman scattering spectroscopic measurements (Figure S5, Table S2).

Figure 1 shows the representative high-pressure raw Brillouin scattering spectra from the two synthesized high-pressure $SiO_2$ phases. These spectra were acquired in two independent series of measurements (Table S1, Figure S6), using centrally positioned samples in the DAC chambers. Although the samples were not annealed, stress conditions were evaluated at the synchrotron for the $CaCl_2$-type and $\alpha$-$PbO_2$-type phases at 92 and 99 GPa, respectively, indicating deviatoric stresses of ~5–6 GPa (Figure S1, S2). To ensure the reliability of Brillouin data under ultrahigh-pressure conditions, we collected sharp, high-quality peaks by measuring multiple angular orientations and extending acquisition times over several days (Fig. 1, Table S1). Since deviatoric stress typically induces peak broadening in Brillouin spectra, the absence of systematic broadening in our data suggests that stress did not progressively increase with pressure. Both the lowest- and highest- pressure data points in Fig. 1 are within ±0.1 of the average FWHM value, consistent with the high-pressure synchrotron data indicating ~5–6 GPa of stress (Fig. 1, Figure S1, S2, Supplementary Text 5). For each Brillouin pressure point (Table S1), pressure was determined using the Raman $T_{2g}$ mode, measured at several spots within the central ~20 μm region of the probed sample both before and after each Brillouin acquisition. The values were averaged, and the standard deviation is reported in Table S1. A cross-check between Raman $T_{2g}$-derived pressures and those obtained from the equation of state (EoS) of the NaCl B2 phase using synchrotron XRD at 92 GPa and 99 GPa (for the $CaCl_2$-type and $\alpha$-$PbO_2$-type $SiO_2$ phases, respectively) showed that Raman-based pressures were up to 2 GPa lower at these high-pressure points. The shear wave velocity profiles from the two phases are shown in Fig. 2 along with previous experimental and theoretical results[24,28,50]. All acquired spectra were subject to background subtraction, which resulted in a consistent $V_S$ reduction within 0.02 km/s in $\alpha$-$PbO_2$-type in all collected data, and within a maximum of 0.2 km/s in the $CaCl_2$-type. The 3rd-order finite strain fits of the shear wave velocity data, shown by continuous lines in Fig. 2 (see Table S2), result in the shear moduli ($G_0$) and its pressure derivatives ($G_0$') as follows: $G_0$ = 180 ( ± 2) GPa, $G_0$' = 1.56 ( ± 0.02) for $CaCl_2$-type $SiO_2$ phase and $G_0$ = 148 ( ± 2) GPa, $G_0$' = 1.67 ( ± 0.01) for $\alpha$-$PbO_2$-type $SiO_2$ phase. Although the slope of the velocity profile of the $\alpha$-$PbO_2$-type $SiO_2$ phase is far steeper than that of the $CaCl_2$-type $SiO_2$ phase, the $\alpha$-$PbO_2$-type $SiO_2$ phase exhibits lower shear velocities than those of the $CaCl_2$-type $SiO_2$ phase throughout the pressure range that we explored (Fig. 2). This could possibly result in ~1.5% negative shear velocity contrast ($\Delta V_S$) under ambient conditions at the expected $SiO_2$ phase transition pressure (120–125 GPa)[45–48] (Figs. 2, 3). This could further potentially increase to $\Delta V_S$ of ~3.0–3.2% under cold slab-[55] or lower mantle geotherm[56] (Fig. 4, Table S6) (see Modeling of $V_S$ profile of MORB for more details).

As shown in Fig. 2, the most significant difference between our results and the prior theoretical studies[24,50] is that our shear wave velocity

$V_S$ values are on average lower by 7–14%. It should be noted that the ferroelastic transition along with the stishovite to the $CaCl_2$-type phase transformation was not observed under the pressure conditions that we explored. This is anticipated from theory to induce a sharp shear softening[24], and was also shown in an X-ray diffraction and Brillouin study of a single-crystal stishovite at 55 GPa by Zhang et al., 2021[44] or polycrystalline stishovite from Brillouin study of Asahara et al. 2013[28] at the pressure range of 25–35 GPa. Starting this study at a relatively high pressure of 55 GPa in polycrystalline $SiO_2$ ($CaCl_2$-type), similar to the material examined by Asahara et al. 2013[28], may explain why the ferroelastic transition - previously observed at 25–35 GPa - was not detected in our work. Comparison with previous experimental results on $CaCl_2$-type $SiO_2$ up to 60 GPa[28], which determined $G_0$ as 179 ( ±3) GPa and $G_0$' as 1.80 ( ± 0.06), shows that the value of $G_0$' in our study is approximately 15% lower. In contrast $G_0$ closely aligns with the previous study[28]. However, extrapolating data from[28] would lead to a remarkable difference in shear wave velocity at the lowermost mantle pressure condition, reaching up to 5% (Fig. 2). The notable difference in $G_0$' may be attributed to the lower pressures examined in the previous study[28], compared to those prevailing at the CMB. Asahara et al. 2013[28] performed only a single velocity measurement at each pressure point, keeping the cell orientation fixed throughout. In contrast, we conducted multiple velocity measurements at each pressure point using different cell orientations. When velocity measurements are conducted at a fixed angle throughout a single series of measurements, the fundamental assumption in polycrystalline measurements - that the sample is a randomly oriented, fine-grained aggregate - may no longer hold. This is especially true if the grain size or crystallographic texture has developed significantly within the sample. In such cases, there is a considerable risk that the measured velocities will deviate substantially from the true average velocity representative of a polycrystalline aggregate. In contrast, to minimize such concerns, we performed measurements at multiple orientations under the same pressure condition. This practice has been routinely adopted in previous studies using Brillouin scattering on polycrystalline samples, and we believe this approach improves the reliability of our measurements compared to those of Asahara et al. 2013[28], whose methodology did not include this standard step. Therefore, the results of Asahara et al. 2013[28] may have been influenced by the development of a preferred crystallographic orientation or texturing, potentially explaining the observed ~0.3 km/s higher average $V_S$ profile over ~30–130 GPa (Fig. 2). The differences are less discernible at lower pressures but become more pronounced in the extended range of ~60–130 GPa. The precise factors responsible for the significant inconsistency with theoretical results remain uncertain. Nonetheless, an important aspect to emphasize is that theoretical predictions for the $G_0$ of $SiO_2$ phase (either stishovite or $CaCl_2$-type) fail to adequately replicate the established value determined through experimental studies conducted under ambient conditions, diverging from it by approximately 18%[24,25]. A similar discrepancy on CaPv between computational[37,57] and recent experimental results was observed[26,27]. Furthermore, a noteworthy disparity arises in estimating the phase transition pressure from $CaCl_2$-type to $\alpha$-$PbO_2$-type, with a 10–20 GPa variation established through X-ray experimental results[45,47,48].

## Modeling of $V_S$ profile of MORB

By combining previously reported thermodynamic parameters[25] (Table S4) with the present shear wave velocity data (Fig. 2, Table S1), we modeled the shear wave velocity profiles in the lowermost mantle. This model considered the potential presence of our experimentally determined dense high-pressure $SiO_2$ phases within the ancient subducted oceanic crust under relevant high-pressure and ambient temperature (Fig. 3). The assessment of potential high-temperature conditions was modeled in Fig. 4, and compared to the 1-dimensional seismic model (PREM)[58] and the theoretically predicted shear wave velocity profiles of MORB[31], which were adopted as reference points to assess the potential contribution of $SiO_2$ phases

(expressed as negative shear wave velocity contrast across discontinuities ($\Delta V_S$)) to the observed negative anomalies ($\delta\ln V_S$) within LLSVPs (see Fig. 4, Figure S7).

For this purpose, we applied two anticipated temperature profiles for the subducting slab and the surrounding lower mantle, respectively[55,56] (Figure S8). The mineral assemblage, molar fractions, and chemical composition in the MORB system under lower mantle conditions were adopted from the previous experimental results[18] (Table S3). All thermo-elastic parameters for the constituent minerals of the MORB used in this modeling are presented in (Tables S4, S5), and are exclusively derived from previous high-pressure experiments[18]. The resultant shear wave velocity profiles of the CaCl$_2$-type and α-PbO$_2$-type SiO$_2$ phases, as well as the MORB, as a

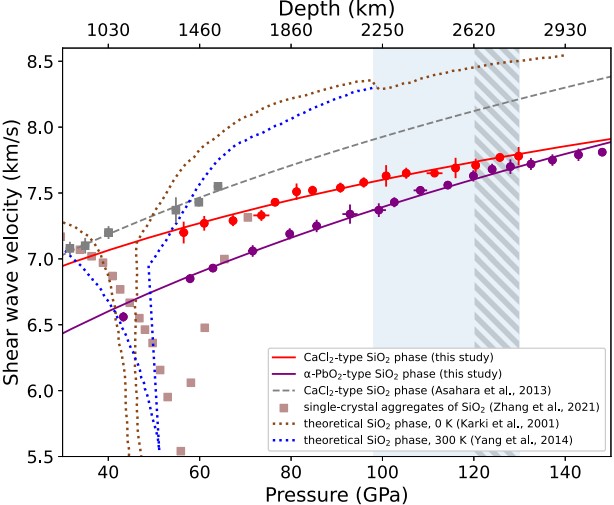

**Fig. 2 | Shear wave velocity profiles of high-pressure SiO$_2$ phases under high-pressures.** Bold lines represent curves fitted using the third-order finite strain equation to experimental data from[48] and $V_S$ data obtained in this study, averaged at each pressure. Error bars indicate uncertainties in pressure and $V_S$. Dotted lines represent the shear wave velocity profiles from the computational studies[24,50]. Dashed line represents the fitted curve of shear wave velocity profile from the experimental study of[28]. The grey shaded area indicates the possible phase transition pressure range from CaCl$_2$-type to α-PbO$_2$-type SiO$_2$ estimated from previous experiments[45–48]. The blue shaded area indicates the expected pressure range where the LLSVPs are primarily observed[5].

function of pressure up to the CMB condition, considering the two distinct temperature conditions, are shown in Fig. 4. We found that the phase transition from CaCl$_2$-type to α-PbO$_2$-type in the MORB and pure SiO$_2$ system could lead to discontinuous shear wave velocity reductions, reaching a maximum of $\Delta V_S$ of 0.6% and 3.2%, respectively (Fig. 4, Table S6). If we assume the modeled MORB under ambient temperature (Fig. 3), the contribution of SiO$_2$ phase transition, as the negative shear velocity contrast across a discontinuity feature in MORB, would decrease to $\Delta V_S$ of ~0.33%. Our experimental results reveal that the CaCl$_2$-α-PbO$_2$ phase transition in SiO$_2$ produces a negative velocity discontinuity. At the transition pressure in pure SiO$_2$, along the cold slab geotherm, this discontinuity corresponds to a decrease of approximately 3% in shear wave velocity. However, when considered in the context of MORB compositions, this reduction is mitigated to about 0.6%, which is relatively minor compared to the overall discrepancy of ~7–14% between the experimentally determined SiO$_2$ velocity profile and theoretical predictions. This highlights the significant role of the overall velocity reduction in SiO$_2$, which has a greater impact on the seismic velocity profile than the phase transition alone. The magnitude of the negative shear wave velocity contrast across a discontinuity ($\Delta V_S$), as determined in this study, closely aligns with the results of previous theoretical calculations[25] (Tables S4, S6). It is known that there is some discrepancy in the elastic properties of cubic-CaPv, as determined in two previous high-pressure experimental studies[26,27]. However, the results from the study[26], characterized by relatively larger $G_0$ and smaller $G_0$' values, than those from the other research[27], effectively counterbalance the velocity differences under high-pressure conditions (Tables S4, S6, Figure S9). As a result, the choice between the two options did not change the final MORB seismic structures, as shown in (Table S6, Figure S9). We also modeled that the expected variations in the bulk moduli ($K_0$) and its pressure derivatives ($K_0$'), as reported in previous experimental studies on SiO$_2$ high-pressure phases[46,48,49,51,54], might not result in the final negative shear velocity contrast ($\Delta V_S$) variations exceeding 2.8–3.6% in the pure SiO$_2$ system and 0.5-0.7% in MORB, respectively (see Tables S3, S6, S7). Assuming that the α-PbO$_2$ phase in MORB reaches an abundance of up to 23 wt.% in the lowermost mantle[18], a more pronounced shear wave velocity reduction of 1.1–1.3% ($\Delta V_S$) in MORB system might be expected (Tables S3, S6, S8).

## Comparison with PREM

Comparative analysis with PREM eventually reveals that the pure SiO$_2$ and the MORB systems might lead to a shear velocity decrease on average of approximately 4% and 7%, respectively, following a cold slab geotherm

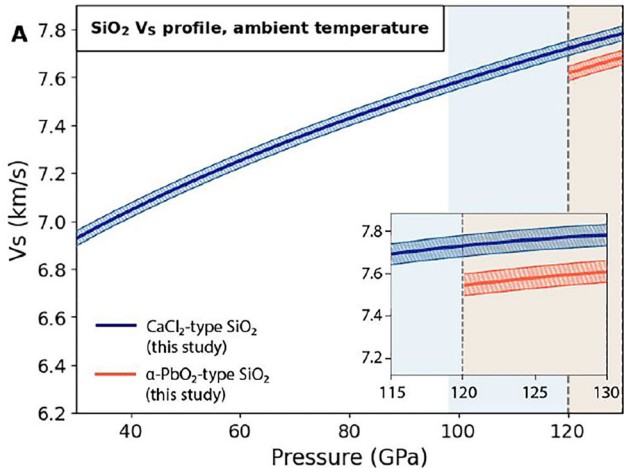

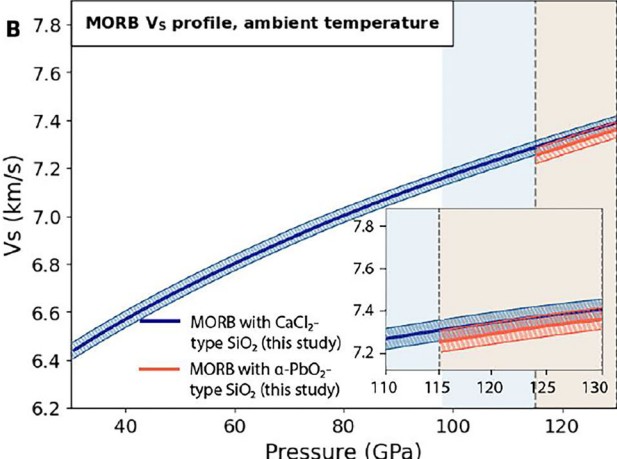

**Fig. 3 | Shear wave velocity profiles of SiO$_2$ and MORB assemblage at ambient temperature. A** Shear wave velocity profiles of SiO$_2$ phase as a function of pressure, assuming a temperature variation of ±100 K. **B** Shear wave velocity profiles of MORB assemblage as a function of pressure, assuming a temperature variation of ±100 K. The adopted MORB composition and its molar fractions can be found in

(Table S3). For CaPv phase, a recent experimentally-derived cubic phase[26] was adopted. The blue shaded area indicates the expected pressure range where the LLSVPs are primarily observed[5]. The orange area represents the stability field range of α-PbO$_2$-type SiO$_2$ phase[45–49].

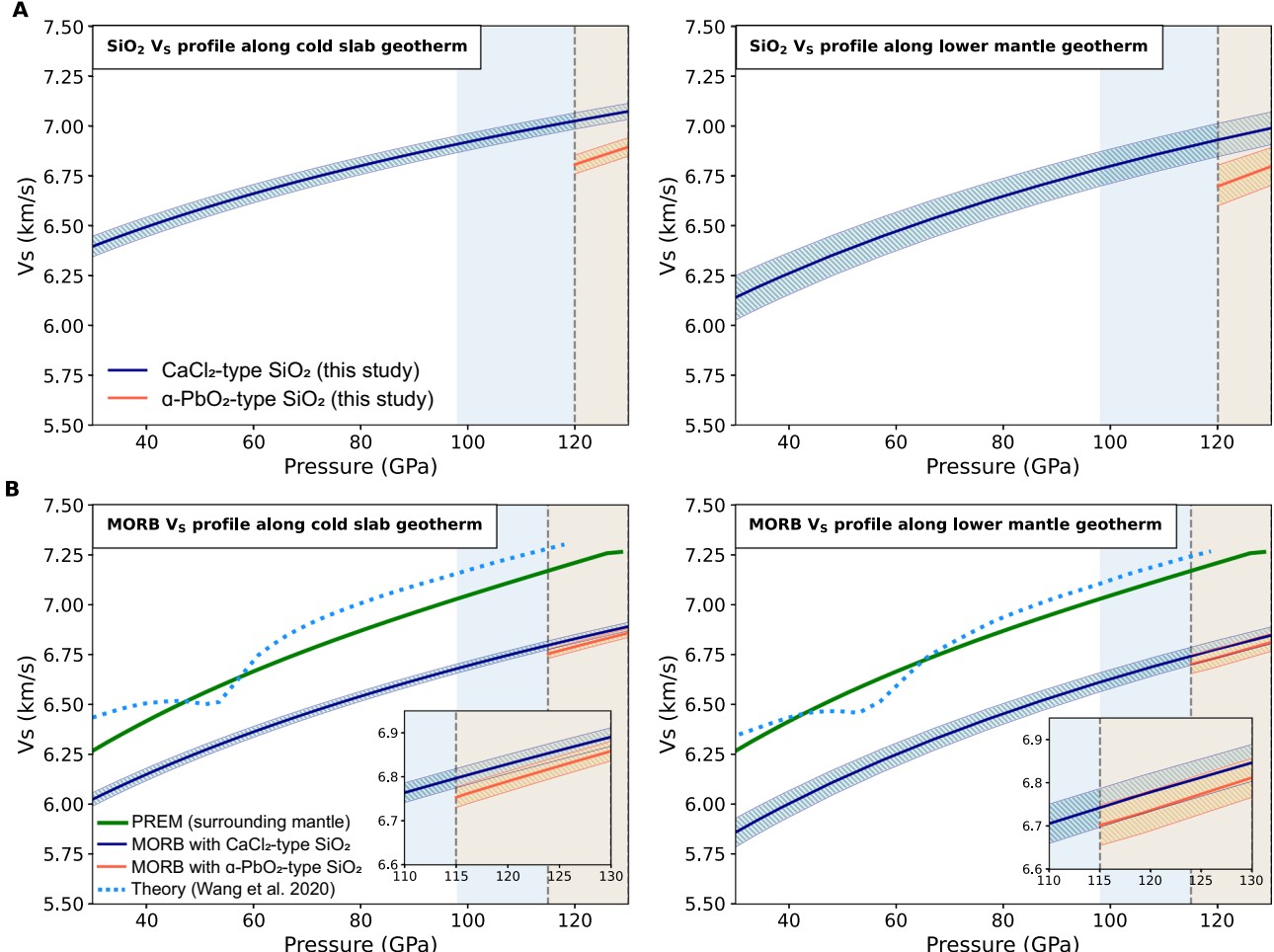

**Fig. 4 | Shear wave velocity profiles of SiO₂ and MORB assemblage under different mantle geotherms. A** Shear wave velocity profiles of $SiO_2$ phase as a function of pressure along with the cold slab geotherm[55] and lower mantle geotherm[56].
**B** Shear wave velocity profiles of MORB assemblage as a function of pressure along with the cold slab geotherm[55] and lower mantle geotherm[56]. In the sequential order, the geotherms consider consistently averaged temperature variations of ±100 K and ±200 K, in comparison to the adopted geotherms used in the theoretical prediction

of MORB $V_S$ profiles[31] (Fig. S7). The adopted MORB composition and its molar fractions can be found in (Table S3). For CaPv phase, a recent experimentally-derived cubic phase[26] was adopted. The comparison of all adopted geotherms and resultant negative shear wave velocity contrast ($\Delta V_S$) (with the expected $SiO_2$ phase transitions) in both systems can be found in (Tables S4-S8). The blue shaded area indicates the expected pressure range where the LLSVPs are primarily observed5. The orange area represents the stability field range of $\alpha$-PbO₂-type $SiO_2$ phase[45–49].

(Fig. 4, Figure S7). On the contrary, as shown in the supplementary materials (Figure S7), considering the results employing the elastic properties of $SiO_2$ phases derived from the theoretical studies[25], both the pure $SiO_2$ system and the MORB system (only in case of following the cold slab geotherm) could exhibit shear velocities higher than those projected by PREM. Given these results, the newly acquired elasticity data of $SiO_2$ high-pressure phases under extreme pressure conditions can offer vital insights into a more comprehensive understanding of the seismic structure of LLSVPs.

## Comparison with theoretical models and experimental data

Theoretical $G_0$ and $G_0$' values tend to be overestimated compared to experimental values, such as in case of CaPv[26,27] or our experimentally determined $SiO_2$ phases in this study. Consequently, using these theoretical parameters in modeled MORB compositions, particularly for dominant phases, can lead to an elevated $V_S$ profile and an overestimated MORB volume fraction required to explain the observed −1.5% anomaly ($\delta lnV_S$). Additionally, discrepancies between studies, such as our work and those by Thomson et al. 2019[27] and Wang et al. 2020[31], may arise from differences in the partitioning behavior and proportions of mineral phases used in MORB modeling. In this study, we assessed the overall velocity profile of MORB based on experimentally determined mineral phases and incorporated

updated constraints on $G_0$ and $G_0$' for all bridgmanite endmembers (Table S4). The lack of high-temperature experimental data on MORB's mineral phases under lower mantle conditions presents challenges in reconciling discrepancies with previous studies. While Mattern et al. 2005[59] emphasized the primary influence of pressure on the $V_S$ profile, the role of $G_0$' remains critical for interpreting seismic observations. Furthermore, previous studies[25,60] suggest that temperature sensitivity in $\eta_{S0}$ can introduce experimental uncertainties of ~10%. In our study, such uncertainties correspond to a ± 1 vol.% variation in MORB content required to reproduce the seismically observed anomalies ($\delta lnV_S$) of −1.5% to −3%, via modeled negative shear wave velocity contrast ($\Delta V_S$) of a similar magnitude.

If the CaPv content in MORB increases by 7 wt.% (from 23 wt.% in Table S3 to 30 wt.%), as suggested by Ricolleau et al. 2010[41], while the $SiO_2$ phase decreases from 17 wt.% to 10 wt.% - a value significantly lower than those reported in other studies e.g., Hirose et al. 2005; Perrillat et al., 2006; Ricolleau et al. 2010; Ishii et al., 2022[18,41,61,62] - then the same MORB volume fraction could still account for the observed seismic anomalies ($\delta lnV_S$). This adjustment would slightly reduce the $SiO_2$ contribution, as the negative shear velocity contrast in MORB, from $\Delta V_S$ of ~0.6% (Fig. 4, Table S6) to $\Delta V_S$ of ~0.5%.

Our study presents a comprehensive analysis of the potential discontinuous feature associated with MORB in the lower mantle (Table S9).

The MORB volume fractions required to explain the observed seismic velocity anomalies are significantly lower than those proposed by previous studies - for example, Thomson et al., 2019[27] estimated up to 64% MORB at 100 GPa and 48% at 125 GPa to explain a $\delta\ln V_S$ of $-1.5\%$. In contrast, our results indicate that 23–33 vol.% MORB may already account for a $\delta\ln V_S$ of $-1.5\%$ (Table S9, Fig. 4). A more pronounced anomaly $\delta\ln V_S$ of -3% would require a higher fraction of ~47–66 vol.%. Importantly, these values decrease under elevated temperatures exceeding 3000 K (Table S9). For instance, if lower mantle temperatures range from 2600 K at the top of the D″ layer to 4000 K at the core-mantle boundary, as suggested by Manthilake et al. 2011[63], the modeled shear velocity contrast ($\Delta V_S$) could increase even by a factor of two (e.g., from ~-0.6% at <2600 K to ~−0.7% at 3000 K and ~ −1% at 4000 K). This increase in $\Delta V_S$ would in turn reduce the required MORB volume fraction - for example, from 33 vol.% under a cold slab geotherm (Case 2 A) to 22 vol.% at 3000 K and as low as 14 vol.% at 4000 K (Table S9).

These results highlight the importance of experimental constraints in refining theoretical models and interpreting seismic profiles in the lower mantle. Although the effect of the negative shear velocity contrast in MORB, with the $SiO_2$ phase transition ($CaCl_2$-type to $\alpha$-$PbO_2$-type) is relatively small ($\Delta V_S$ of ~0.6–1%, Tables S6– S8) under assumed cold slab or lower mantle geotherms - and further diminishes to $\Delta V_S$ of ~0.33% under assumed ambient conditions (Fig. 3) - the collective contribution of MORB phases with experimentally refined $G_0'$ values provide valuable insight into how a realistic $V_S$ profile of MORB can decrease relative to the PREM model. This study contributes to understanding the potential role of MORB in explaining the negative anomalies ($\delta\ln V_S$) in the range of -(1.5–3)%.

## Origin of LLSVPs
It is generally accepted from previous shear wave tomographic observations that the negative shear velocity anomalies in LLSVPs exhibit depth-dependent variations with anomalies ranging from approximately −0.5% to −1% in shallow regions[3,5,7,8] to around −3% at depths ranging from 100 to 200 km from the CMB[6,12,14] (Fig. 5). If such depth-dependent shear velocity anomalies are simply attributed to the contamination of MORB lithology into the surrounding mantle, near the bottom of the lowermost mantle, and without relying on temperature anomalies, a −1.5% anomaly ($\delta\ln V_S$) within the LLSVP's stability field, where $SiO_2$ phase undergoes a transition from $CaCl_2$-type to $\alpha$-$PbO_2$-type, may be explained by the presence of approximately ~23–33 vol.% MORB (Fig.4, Tables S6–S9). Assuming that the $CaCl_2$-$\alpha$-$PbO_2$-type phase transition leads to silica enrichment in the lower mantle[18] (Tables S8, S9), 22 vol.% of MORB could be sufficient to account for a −1.5% ($\delta\ln V_S$) anomaly in a cold slab. Instead, the distinctive $\delta\ln V_S$ of

−3% observed specifically at the bottom of the mantle could be explained by the combined effect of temperature effect from cold slab or lower mantle geotherms, phase transition of $CaCl_2$- to $\alpha$-$PbO_2$-type phase of $SiO_2$ in the accumulated MORB pile, and/or variation of MORB pile volume fraction (Tables S7–S9). Since the exact temperature around the CMB has not been conclusively established, it is suggested that the temperature near the base of the mantle deviates significantly from the adiabatic temperature gradient, possibly reaching around 3000 K to 4000 K[64–66]. For instance, if we assume a temperature of ~3890–3900 K at the very bottom of the lower mantle, the observed shear wave velocity anomaly ($\delta\ln V_S$) of −3% can be reasonably attributed to the presence of ~21–28 vol.% MORB (Table S9).

Depending on the temperature conditions at the CMB, it is possible that just above the CMB, temperature surpasses the solidus of MORB[64,67], and contributes to the potential formation of partial melt. These factors might be significant contributors to the formation of the Ultra-Low Velocity Zones (ULVZs) at the CMB[65], but have not been investigated in more detail in this study. Instead, this study focused on the combination of factors such as temperature, an abundance of $\alpha$-$PbO_2$ phase in MORB (up to 23 wt.% in the lowermost mantle), and the MORB volume fractions variations on the seismic structure of LLSVPs. The results of this analysis, presented in (Table S9), suggest that the temperature effect of ~3000–4000 K gains significance in generating a larger anomaly, such as −3% ($\delta\ln V_S$), and thus lowering the vol.% of MORB below 44%. This could be attributed to the unique high-temperature conditions near the CMB. Having considered all factors, it is also noteworthy that in the mineral assemblage of both MORB and the ambient mantle, the post-perovskite phase transition could occur at around 113–120 GPa[3,18,68]. Despite the fact, that post-perovskite can potentially form within a close range to the examined $SiO_2$ phase transition in this study, it is also well known that the shear velocity jump across the post-perovskite phase transition strongly depends on its texture development[69,70]. Thus, in case we assume an elastically isotropic condition, no significant positive jump in shear velocity $V_S$ should be expected[68]. If detected, however, the effect of the post-perovskite phase transition might not be that significant. Therefore, the present study demonstrates that the lower magnitude of LLSVPs' anomaly ($\delta\ln V_S$) (−1.5%) can be explained by the presence of only around ~23–33 vol.% of MORB, without relying on thermal anomalies (Fig. 5, Table S9). In addition to this, the discrepancy between theoretically-predicted[24,50] and experimentally-derived shear velocities in $SiO_2$ in this study was revealed.

Among the subduction process hypothesis, proposing that chemical heterogeneity is a primary cause of LLSVPs[3,15–17], there are alternative explanations for such a cause. One of them suggests that they could be attributed to primordial residues inherited from early differentiation

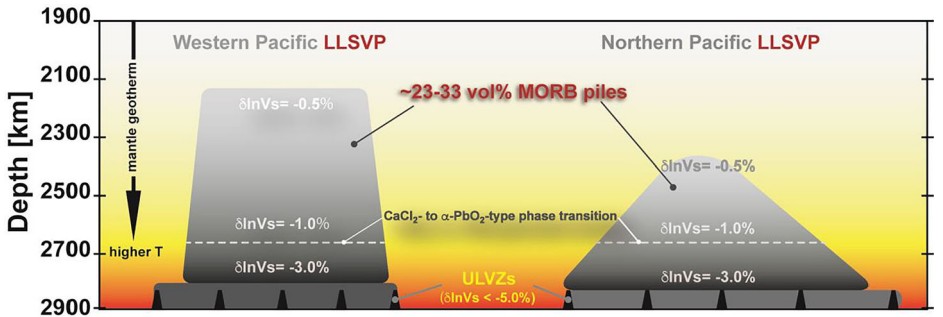

**Fig. 5 | Conceptual schematic of Pacific LLSVPs and related features.** The negative shear velocity anomalies ($\delta\ln V_S$) in the LLSVPs exhibit depth-dependent variations with anomalies ranging from approximately −0.5% to −1% in shallow regions[3,5,7,8] and around -3% at the base (bottom 100–200 km)[6,12,14]. The present study demonstrates that averaged $\delta\ln V_S$ typically observed at LLSVPs (−1.5%) can be explained by the presence of ~23–33 vol.% of MORB, with a discontinuous feature across $SiO_2$ phase transition, and without relying on thermal anomalies (+ 1500 K). The distinctive $\delta\ln V_S$ of −3% observed specifically at the bottom of the mantle could be explained by the combined effect of high-temperature, phase transition of $CaCl_2$- to $\alpha$-$PbO_2$ phase of $SiO_2$ in the accumulated MORB pile, and/or variation of MORB pile volume fraction. This conceptual representation is based on interpretations discussed in He & Wen (2009, 2012)[11,90], including features developed in this work.

**Article**

processes in the Earth. This hypothesis proposes that the low shear velocity anomalies within LLSVPs are caused by high-density phases enriched in iron that selectively crystallized from the basal magma ocean in early Earth[71,72]. This hypothesis can also account for the low shear velocity and high-density features of LLSVPs. While we cannot disregard the importance of the primordial heterogeneity hypothesis proposed by several geodynamic studies[2,8,20,73,74], it remains unclear whether a sufficient amount of primordial residues near the CMB can be created or preserved to account for the size of LLSVPs. Another, more recent hypothesis suggests that LLSVPs formed through a giant-impact scenario involving Theia colliding with proto-Earth[75]. This scenario involves entraining both molten and solidified fractions, which were subsequently sunk into the LLSVPs regions. These two hypotheses, however make it challenging to provide a strong inherent explanation for why LLSVPs are specifically located beneath Africa and the Pacific. On the other hand, considering at least 2 billion years of the history of subduction processes[76] into the Earth's interior, the presence of ~23–33 vol% of oceanic crust within LLSVPs appears to be a feasible accumulation (Table S9)[77,78].

The findings of this study, which demonstrate that relatively small amounts of oceanic crust can explain seismological characteristics of LLSVPs, it is worth noting that previous research has also highlighted the greater density of oceanic crust compared to the surrounding mantle, which could maintain seismic heterogeneity persistently[18,40]. Considering these factors collectively, this hypothesis seems to be a suitable explanation for the origin of LLSVPs.

Furthermore, the presence of a large-scale, compositionally distinct basaltic pile with a laterally skewed distribution directly above the core-mantle boundary may introduce lateral heterogeneity to the heat transport mechanism from the core[2,74,79]. This may imply the possibility of receiving an enhanced thermal flux in certain regions, although without creating abrupt thermal anomalies of $+1500\,K$, and potentially lead to significant mantle upwelling flows. The heterogeneity in heat transport mechanism is expected to partially contribute, along with the process of subducted MORB material, to the shear velocity reduction in LLSVPs at the very bottom region.

The long-term accumulation of the chemically distinct subducted oceanic crust in the lowermost mantle is thus anticipated to alter the bulk chemistry of the lower mantle towards more $SiO_2$- and $Al_2O_3$-rich composition over the subduction history. The effect of $Al_2O_3$ across stishovite to $CaCl_2$-type $SiO_2$ phase transition was shown to reduce the transition pressure (e.g., refs. 80–83) and to decrease the bulk modulus ($K$) of $SiO_2$ phases (e.g., see refs. 48,84–87). However, the shear properties of Al-bearing $CaCl_2$-type or $\alpha$-$PbO_2$-type $SiO_2$ phases remain experimentally unconstrained. For example, Lakshtanov et al. 2007[85] investigated Al- and H-bearing stishovite at room pressure up to ~25 GPa and reported a decrease in shear modulus ($G$) compared to pure room pressure $SiO_2$ (e.g., Jiang et al. 2009[88]). These results, however, are limited to the stishovite stability field and involve a minor presence of water. The current absence of experimental data on the shear modulus ($G$) and its pressure derivative ($G'$) for Al-bearing $CaCl_2$-type and $\alpha$-$PbO_2$-type $SiO_2$ phases, particularly under anhydrous conditions, highlights the importance of further high-pressure elasticity studies to quantify the effect of $Al_2O_3$ on the seismic properties of these high-pressure $SiO_2$ phases.

## Reporting summary
Further information on research design is available in the Nature Portfolio Reporting Summary linked to this article.

## Data availability
All data supporting the findings of this study are available in the main text and Supplementary Information.

## Code availability
No custom code was developed for this study. Data analyses were carried out using OriginPro. 2023, Python 3.10 (including NumPy, SciPy, and Matplotlib), and the open-source BurnMan. package (version 1.1.0). Dioptas

and PDIndexer were used for X-ray diffraction data integration, and indexing. All input parameters necessary to reproduce the results are provided in the main text and Supplementary Information.

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

## Acknowledgements

This research was supported by ETH Zurich start-up funding (PSP1-001828-000) and the Swiss National Science Foundation (Grant No. 200021_197187) awarded to Motohiko Murakami. Synchrotron X-ray diffraction experiments were performed at the European Synchrotron. Radiation Facility (ESRF), beamline ID27, Grenoble, France. We thank E. Ito (sample synthesis using large volume press), T. de Selva-Dewint (Brillouin scattering measurements), R. Popa, M. Lüder (FT-IR measurements), N. Ma, S. Merkel, T. Poreba, M. Mezouar (synchrotron XRD measurements) for their experimental assistance. Y. Mori is acknowledged for his valuable comments and suggestions.

## Author contributions

Conceptualization: M.M.; Methodology: E.K., M.M.; Investigation: E.K., M.M., P.S., C.L.; Visualization: E.K., M.M.; Funding acquisition: M.M.; Supervision: M.M., P.S., C.L.; Writing – original draft: E.K., M.M.; Writing – review & editing: E.K., M.M., P.S. and C.L.

## Competing interests

The authors declare no competing interests.
