## [Transparent Peer Review file · Communications Earth & Environment]

Subducted oceanic crust, containing high-pressure SiO₂ phases, contributes to seismic anomalies in LLSVPs

Corresponding Author: Dr Ewa Krymarys

Version 0:

Decision Letter:

Dear Ms Krymarys,

Your manuscript titled "Ultrahigh-Pressure Sound Velocities of Dense SiO₂ Phases and the Origin of the LLSVPs" has now been seen by 3 reviewers, whose comments are appended below. You will see that they find your work of interest. However, they have raised substantial concerns that must be addressed. In light of these comments, we cannot accept the manuscript for publication, but would be interested in considering a revised version that fully addresses these concerns.

We hope you will find the reviewers' comments useful as you decide how to proceed. Should additional work allow you to address these criticisms, we would be happy to look at a substantially revised manuscript. If you choose to take up this option, please either highlight all changes in the manuscript text file, or provide a list of the changes to the manuscript with your responses to the reviewers.

In the revised manuscript, please make sure to:

- Provide compelling arguments that ensure the quality of the reported data/results and explain potential discrepancies with previous studies;
- Clarify the distinction between the two uses of δ In VS and emphasize the comparison of the MORB profile with the 1D average velocity, highlighting the implications of the lower VS compared with prior results;
- Explicitly discuss the influence of CaPv data on MORB fraction estimates.

When resubmitting, please provide a point-by-point response to the reviewers' comments. Please submit your responses as a separate file, distinct from your cover letter where you can add responses to the Editors' comments that you do not want to be made available to the reviewers. Word files are preferred. We recommend that any figures, tables or graphs that are included in the response to reviewers are also included in the main article or Supplementary Information.

If the revision process takes significantly longer than three months, we will be happy to reconsider your paper at a later date, as long as nothing similar has been accepted for publication at Communications Earth & Environment or published elsewhere in the meantime.

Please use the following link to submit your revised manuscript, point-by-point response to the reviewers' comments with a list of your changes to the manuscript text (which should be in a separate document to any cover letter), a tracked-changes version of the manuscript (as a PDF file) and any completed checklist:

Link Redacted

Please do not hesitate to contact us if you have any questions or would like to discuss the required revisions further. Thank you for the opportunity to review your work.

Best regards,

Joao Duarte, PhD
Editorial Board Member
Communications Earth & Environment
orcid.org/0000-0001-7505-3690

Carolina Ortiz Guerrero, PhD
Associate Editor
Communications Earth & Environment

EDITORIAL POLICIES AND FORMAT

If you decide to resubmit your paper, please ensure that your manuscript complies with our editorial policies and complete and upload the checklist below as a Related Manuscript file type with the revised article:

Editorial Policy Policy requirements
(Download the link to your computer as a PDF.)

- Behavioural and social science
- Ecological, evolutionary & environmental sciences
- Life sciences

<https://www.nature.com/documents/nr-reporting-summary.zip>

For your information, you can find some guidance regarding format requirements summarized on the following checklist: (<https://www.nature.com/documents/commsj-phys-style-formatting-checklist-article.pdf>) and formatting guide (<https://www.nature.com/documents/commsj-phys-style-formatting-guide-accept.pdf>).

REVIEWER COMMENTS:

Reviewer #1 (Remarks to the Author):

Comments in the attached pdf file.

Reviewer #2 (Remarks to the Author):

The authors present Brillouin light scattering results of polycrystalline SiO₂ phases at high pressure. The Brillouin results are used to derive V_s of the SiO₂ phases and modelled thermo-elastically at high P-T. The results are applied to understand large low shear velocity provinces in the deeper lower mantle. There are a number of technical and scientific issues that would prevent the work from being published.

These issues are discussed in more details below. Firstly, polycrystalline Brillouin data does not truly represent aggregate V_s. Brillouin spectral intensity is a function of crystallographic orientation and thus is an assemble of all scattered signals from the compressed sample. In this study, no pressure medium was used even though the authors claim that NaCl is used, but it's not a good medium unless it's annealed. Stress and textures can be developed in this type of samples that would further complicate the issue. The interpretation of the spectra is thus a complex issue. Just look at widths and shapes of the spectra. There are some limited studies in the field on this topic, but this issue is not settled as it is sample dependent. Secondly, pressure was not determined reliably. The authors used Raman edge which is mean to be a secondary option (when there's no other option available for P determination). Diamond Raman edge can be influenced by stress environment and diamond itself, and has lots of uncertainties to it. There's a precision vs accuracy issue in what exactly the pressure and its uncertainty the authors are reporting. The authors could have used XRD of their samples for P determination, but they did not. Thirdly, literature data for stishovite and post-stishovite phases are not compared properly. There are very high-quality data and even single crystal Cij data in the literature that are not even cited and discussed in Figure 2 and relevant sections.

It's not clear why the authors did not even do so but this reads like a very bad practice in science. Fourthly, silica phases always contain some Al so Al effects on elasticity should be considered. In fact, there are also rich literature on this issue that was totally ignored in this paper. Lastly, the elasticity and thermodynamic modelling should incorporate full datasets including Vs, Vp, and density. There're no temperature effects measured and addressed here (no new data to address this issue). Based on what we have in the literature, HT effect at HP appears to be a major uncertainty, rather than the data the authors reported. Given all these considerations, there are just huge error bars that would mask out all effects and applications that the authors are trying to address. Figure 3 does not even show uncertainties? At the end, the paper has a nice story to tell but without much of scientific support.

Reviewer #3 (Remarks to the Author):

In this manuscript, Krymarys et al. present new experimental measurements of the shear-wave velocity (VS) of two high-pressure silica phases at pressures up to the core-mantle boundary (CMB). The reported VS values for these silica phases are significantly lower (by approximately 7–14%) than previous theoretical predictions from first-principles calculations. Utilizing these measurements, the authors model the VS profile of mid-ocean ridge basalt (MORB) assuming varying silica contents. Their results indicate a velocity reduction of 0.6% or 1.1–1.3% due to the phase transition from CaCl-type to α -PbO-type silica. Furthermore, they find that the VS of MORB, following a cold slab geotherm, is approximately 7% lower than the PREM model. Consequently, they estimate that a MORB fraction of ~23–33% is required to explain the observed -1.5% low-velocity anomalies within LLSVPs. Based on these results, the authors propose that ancient subducted MORB, even without significant thermal anomalies, may account for LLSVP characteristics.

The methodological approach and workflow are clear, and the new findings contribute significantly to the ongoing debate regarding the origin of LLSVPs. Nonetheless, I recommend acceptance upon major revisions addressing the following substantive issues to enhance the clarity, precision, and impact of this contribution. Please refer to the attached file for detailed reviews.

Communications Earth & Environment is committed to improving transparency in authorship. As part of our efforts in this direction, we are now requesting that all authors identified as 'corresponding author' create and link their Open Researcher and Contributor Identifier (ORCID) with their account on the Manuscript Tracking System prior to acceptance. ORCID helps the scientific community achieve unambiguous attribution of all scholarly contributions. You can create and link your ORCID from the home page of the Manuscript Tracking System by clicking on 'Modify my Springer Nature account' and following the instructions in the link below. Please also inform all co-authors that they can add their ORCIDs to their accounts and that they must do so prior to acceptance.

Version 1:

Decision Letter:

Dear Ms Krymarys,

Your manuscript titled "Ultrahigh-Pressure Sound Velocities of Dense SiO₂ Phases and the Origin of the LLSVPs" has now been seen by the original reviewers #1 and #3, whose comments are appended below. As you will see, while they recognise that some improvements were made, they both agree that several of the points previously raised have not been adequately handled, in particular concerning how they were introduced (or not) in the main text. In light of these comments and the fact that our thresholds were not fully met, we cannot accept the manuscript for publication, but would be interested in considering a revised version that fully addresses the concerns raised by the reviewers.

In the Revised version please make sure to:

- Improve the manuscript by introducing all the changes correctly into the main text, clarifying all the inconsistencies, and improving the narrative of the main text following the suggestions of Reviewer #1.

- Provide compelling arguments that ensure the quality of the reported results and objectively explain potential discrepancies with previous studies and limitations of your approach.

- Make sure to clarify the use of $\delta\ln V_s$, which seems to be being used to denote two different concepts (lateral shear wave velocity anomalies of LLSVPs and the velocity decrease produced by the CaCl₂ type SiO₂ → seifertite phase transition), as pointed out by Reviewer #3.

We hope you will find the reviewers' comments useful as you decide how to proceed. Should additional work allow you to address these criticisms, we would be happy to look at a substantially revised manuscript. If you choose to take up this option, please either highlight all changes in the manuscript text file, or provide a list of the changes to the manuscript with your responses to the reviewers.

When resubmitting, please provide a point-by-point response to the reviewers' comments. Please submit your responses as a separate file, distinct from your cover letter where you can add responses to the Editors' comments that you do not want to be made available to the reviewers. Word files are preferred. We recommend that any figures, tables or graphs that are included in the response to reviewers are also included in the main article or Supplementary Information.

If the revision process takes significantly longer than three months, we will be happy to reconsider your paper at a later date, as long as nothing similar has been accepted for publication at Communications Earth & Environment or published elsewhere in the meantime.

Please use the following link to submit your revised manuscript, point-by-point response to the reviewers' comments with a list of your changes to the manuscript text (which should be in a separate document to any cover letter), a tracked-changes version of the manuscript (as a PDF file) and any completed checklist:

Link Redacted

Please do not hesitate to contact us if you have any questions or would like to discuss the required revisions further. Thank you for the opportunity to review your work.

Best regards,

Joao Duarte, PhD
Editorial Board Member
Communications Earth & Environment
orcid.org/0000-0001-7505-3690

Carolina Ortiz Guerrero, PhD
Associate Editor
Communications Earth & Environment

EDITORIAL POLICIES AND FORMAT

If you decide to resubmit your paper, please ensure that your manuscript complies with our editorial policies and complete and upload the checklist below as a Related Manuscript file type with the revised article:

Editorial Policy Policy requirements
(Download the link to your computer as a PDF.)

- Behavioural and social science
- Ecological, evolutionary & environmental sciences
- Life sciences

An updated and completed version of our Reporting Summary must be uploaded with the revised manuscript
You can download the form here:

<https://www.nature.com/documents/nr-reporting-summary.zip>

For your information, you can find some guidance regarding format requirements summarized on the following checklist: (<https://www.nature.com/documents/commsj-phys-style-formatting-checklist-article.pdf>) and formatting guide (<https://www.nature.com/documents/commsj-phys-style-formatting-guide-accept.pdf>).

REVIEWER COMMENTS:

Reviewer #1 (Remarks to the Author):

[See attachment].

Reviewer #3 (Remarks to the Author):

After carefully evaluating the authors' rebuttal and the revised manuscript, I find that my principal concern has not been satisfactorily addressed. Consequently, I am unable to recommend publication in its present form until this issue is fully resolved. I therefore restate my primary concern below to ensure complete clarity.

In the revised manuscript, the authors continue to use $\delta \ln V_s$ to denote both the lateral shear wave velocity anomalies of LLSVPs and the velocity decrease produced by the CaCl_2 type $\text{SiO}_2 \rightarrow$ seifertite phase transition, which makes the interpretation confusing. They are different concepts and are used inappropriately. For example, in lines 229–232 the authors state, "It is generally accepted from previous shear wave tomographic observations that the negative shear velocity anomalies in LLSVPs exhibit depth-dependent variations with anomalies ranging from approximately -0.5% to -1% in shallow regions to around -3% at depths ranging from 100 to 200 km from the CMB." These values represent velocity anomalies of LLSVPs relative to reference models such as PREM. The authors use the term "anomaly" in line 235 to describe the -1.5 % anomaly correctly; however, in lines 239, 247, 257, and 267 they use "discontinuity" to refer to the same velocity anomalies. These inconsistencies suggest that the authors remain unclear about the distinction between velocity anomalies ($\delta \ln V_s$) and the velocity contrast across a discontinuity (for which I recommend an alternative notation, such as ΔV_s , to distinguish two concepts).

On the other hand, the ΔV_s generated by the phase transition does not play a significant role in explaining the velocity anomaly ($\delta \ln V_s$) in LLSVP based on their results, because its effect in MORB is only ~0.6% without SiO_2 enrichment (Table S6-S7), and its contribution is further decreased when considering the MORB fraction. To illustrate the limited contribution, I recommend adding a representative V_s profile of the assemblage containing 23 vol % MORB (choose one case from Table S9) to Figure 4; In this case, the expected differences caused by the phase transition should be negligible.

However, this does not undermine the importance of this work to a large extent. The main contribution of this work is that the measured V_s is significantly lower than previous estimates, although the high-temperature extrapolation introduces some uncertainty. Therefore, the authors should focus on the velocity profiles with varying MORB fractions relative to PREM, i.e. results in Table S9, which is more important to explain the depth-dependent velocity anomalies from -1.0% to -3.0%. According to Table S9, the change in velocity anomalies in LLSVP primarily results from the effect of temperature and MORB fraction, rather than the phase transition. And the estimated required MORB fractions based on their data should be less than those based on previous data.

Issues with the citations still exist. For example, as I noted in my previous review, Yang et al. (2014) does not contain information on the CaCl_2 type $\text{SiO}_2 \rightarrow$ seifertite transition, whereas Karki et al. (2001) does, but the revised manuscript cites Yang et al. (2014) rather than Karki et al. (2001) at line 99. The authors need to examine every reference carefully to ensure each citation is accurate and appropriate.

Communications Earth & Environment is committed to improving transparency in authorship. As part of our efforts in this direction, we are now requesting that all authors identified as 'corresponding author' create and link their Open Researcher and Contributor Identifier (ORCID) with their account on the Manuscript Tracking System prior to acceptance. ORCID helps the scientific community achieve unambiguous attribution of all scholarly contributions. You can create and link your ORCID from the home page of the Manuscript Tracking System by clicking on 'Modify my Springer Nature account' and following the instructions in the link below. Please also inform all co-authors that they can add their ORCIDs to their accounts and that they must do so prior to acceptance.

Version 2:

Decision Letter:

Dear Dr Krymarys,

Your manuscript titled "Ultrahigh-Pressure Sound Velocities of Dense SiO₂ Phases and the Origin of the LLSVPs" has now been seen by our reviewers, whose comments appear below. In light of their advice we are delighted to say that we are happy, in principle, to publish a suitably revised version in Communications Earth & Environment.

We therefore invite you to revise your paper one last time to address the remaining concerns of our reviewers. At the same time we ask that you edit your manuscript to comply with our format requirements and to maximise the accessibility and therefore the impact of your work.

EDITORIAL REQUESTS:

****Please take care to match our formatting and policy requirements. We will check revised manuscript and return manuscripts that do not comply. Such requests will lead to delays. ****

SUBMISSION INFORMATION:

OPEN ACCESS:

Communications Earth & Environment is a fully open access journal. Articles are made freely accessible on publication. For further information about article processing charges, open access funding, and advice and support from Nature Portfolio, please visit <https://www.nature.com/commsenv/open-access>

Link Redacted

Best regards,

João Duarte, PhD
Editorial Board Member
Communications Earth & Environment

Martina Grecequet, PhD
Senior Editor,
Communications Earth & Environment
Consulting Editor
Communications Sustainability

REVIEWERS' COMMENTS:

Reviewer #1 (Remarks to the Author):

Overall, I am satisfied with the corrections implemented in the manuscript.

There is a last imprecision that should be corrected in the discussion of the effect of Al₂O₃ on the elasticity of SiO₂-stishovite. Ref. 85 is inappropriate here because it is a room pressure study of the effect of Al³⁺ on the elasticity of SiO₂-stishovite. It is not clear what is the 10 GPa range the authors refer to and therefore the statement and the discussion should be modified for correctness.

Reviewer #3 (Remarks to the Author):

All points raised in my earlier review have been fully resolved. I now recommend the manuscript for acceptance in its present form.

** Visit Nature Portfolio's author and referees' website at www.nature.com/authors for information about policies, services and author benefits**

‘Ultra-high-Pressure Sound Velocities of Dense SiO₂ Phases and the Origin of the LLSVPs’

by Krymarys et al.

The manuscript submitted by Krymarys et al. report unique sound velocity measurements on ultrahigh pressure polymorphs of SiO₂, CaCl₂- and α -PbO₂- structured SiO₂) to 150 GPa at room temperature by Brillouin scattering spectroscopy. The results reveal a drop in velocities across the CaCl₂- to PbO₂ structured SiO₂ that further contribute to decrease the velocities of subducted MORB at the bottom of the mantle. Based on this observation, together with modelling of MORB velocities along different thermal regimes (slab vs. mantle geotherm), the authors propose that the Large Low Shear Velocity Provinces (LLVSP) at the bottom of the mantle could be explained by the presence of ~23-33% subducted oceanic crust.

The topic is highly relevant and of potential interest for the readership of COMMSENV. The paper is well crafted and the figures and supported materials are appropriate. However, despite the novelty of the data, the conclusions do not appear supported by the reported data and may rather results from the modelling and associated assumptions. There are also some unexplained inconsistencies between the reported datasets and the previous studies (some of them also ignored here) that have not been addressed in the manuscript as I point out in the detailed comments below. Moreover, the data has not been made available, which difficult the verification of the claims. Based on these concerns, I cannot recommend the acceptance of the manuscript in the present form.

Below there is a list of the major and minor concerns about the data, methodology and implications:

- 1) The paper does not seem to give proper credit to previous experimental studies of the high-pressure elasticity of SiO₂ polymorphs stishovite and CaCl₂-structured phase. I am referring here to the single-crystal elasticity data of Zhang et al. (PRL 2021) up to 70 GPa, which should be properly acknowledged in the introduction and included in the comparison reported in Fig.2 (see minor comments below). It is not clear why these valuable data have been excluded and the comparison is restricted to previous own data and theoretical calculation. This is not a very scholarly approach.
- 2) The authors emphasize that their results for CaCl₂-phase reveal velocities that are substantially lower than predicted by theory, which would support the potential role of subducted MORB in explaining the negative velocity anomalies in the LLSVPs. However, they appear to avoid the comparison with previous own data on polycrystalline CaCl₂-SiO₂ (Ref. 28, Asahara et al. 2013), which are substantially higher than those reported in the present study. The comparison is reduced to the extrapolated velocities, which depend on the fitted shear elastic moduli and pressure derivatives as claimed by the authors (lines 144-148). However, the direct comparison of the velocities in the pressure range of overlap (around 60 GPa) shows differences which are well beyond the uncertainties in shear velocities reported (Suppl. Table S1). How can the authors explain the inconsistencies with previous own measurements where the experimental protocols would be assumed as homogenous? This should be discussed in the manuscript rather than ignored because the disagreement is obvious in Fig.2 and the lack of comparison makes the results suspicious.
- 3) Following in the previous comment, the only possibilities to explain the disagreement is the quality of the polycrystalline SiO₂ samples (compaction degree, textures, etc) employed here

and in Asahara et al. (2013), or by the fact that the present data may suffer from vignetting (Sinogekin et al., 1998), which will result in systematically lower velocities. Therefore, until a reasonable explanation for the inconsistencies in own measurements is not provided, the data reported here is difficult to trust. At least in single-crystal studies there are several examples of interlaboratory consistency in the sound velocity data, and this should be also expected for polycrystalline samples if the experiments are properly conducted.

- 4) Looking at the velocity data for CaCl₂ and α -PbO₂ at the bottom of the mantle (Fig.2), the velocity contrast is almost unresolvable within mutual uncertainties so the authors should not be putting that much weight on the contrast. Also, it is not clear in the text (lines 133-134) in the reported contrast is derived from the present study or from the cited references. Please clarify. Given the unresolvable contrast, it is not clear to me how this can result in an enhanced contrast along the slab and mantle geotherms as shown in Fig.3, unless this results from the parameters and assumptions adopted in the projection of velocities along the geotherm. This should be clarified.

Minor comments:

Line 90: Reference Zhang et al. 2021 (*Physical Review Letters*, 126, 025701), which has the added value of being single-crystal elasticity data up to 70 GPa, should be cited here.

Line 114-116: Sentence should be reconsidered for clarity: 'The CaCl₂-type SiO₂ phase was synthesized in a DAC upon compression above 55 GPa using stishovite, which was pre-synthesized using a large-volume press, as starting material (or similar).'

Line 139-140: Softening has been also confirmed by single-crystal Brillouin scattering. Ref. Zhang et al. PRL 2021 should be included in the discussion here.

Line 198: should read: '...under extreme pressure conditions should...'

Figure 2: The aggregate shear wave velocity data from the single-crystal study of Zhang et al. (PRL 2021) should be included in the comparison. All necessary data for the comparison is available with the manuscript. Also, the lack of consistency with previous experimental studies (Ref. 28) should not be ignored.

In summary, the authors should provide compelling arguments to ensure the quality of the data reported here and to explain the differences with previous (own) studies. And the fact that velocity contrast appears resolvable when projected along the geotherms when it is not a clear feature in the original room temperature data.

In this manuscript, Krymarys et al. present new experimental measurements of the shear-wave velocity (V_S) of two high-pressure silica phases at pressures up to the core-mantle boundary (CMB). The reported V_S values for these silica phases are significantly lower (by approximately 7–14%) than previous theoretical predictions from first-principles calculations. Utilizing these measurements, the authors model the V_S profile of mid-ocean ridge basalt (MORB) assuming varying silica contents. Their results indicate a velocity reduction of 0.6% or 1.1–1.3% due to the phase transition from CaCl-type to α -PbO-type silica. Furthermore, they find that the V_S of MORB, following a cold slab geotherm, is approximately 7% lower than the PREM model. Consequently, they estimate that a MORB fraction of ~ 23 –33% is required to explain the observed -1.5% low-velocity anomalies within LLSVPs. Based on these results, the authors propose that ancient subducted MORB, even without significant thermal anomalies, may account for LLSVP characteristics.

The methodological approach and workflow are clear, and the new findings contribute significantly to the ongoing debate regarding the origin of LLSVPs. Nonetheless, I recommend acceptance conditional upon major revisions addressing the following substantive issues to enhance the clarity, precision, and impact of this contribution.

1 Misuse of $\delta \ln V_S$

The manuscript uses $\delta \ln V_S$ to describe both the velocity anomalies within LLSVPs (e.g., lines 33, 212) and the negative velocity change (e.g., 177) associated with phase transitions. This dual usage may confuse readers. For example, line 208 incorrectly refers to LLSVP velocity anomalies as a “discontinuity”. The -1.5% $\delta \ln V_S$ represents deviations relative to an average reference velocity at the same depth, whereas a discontinuity denotes a velocity change across a phase transition depth. The authors should clearly distinguish these concepts throughout the text.

2 Focus on the comparison of MORB profile with 1D reference model

The velocity reduction caused by the phase transition is minor, approximately 1% or less in MORB, and decreases further when considering MORB fractions within the mantle. Consequently, the phase transition’s effect on the overall velocity profile is negligible. In my opinion, the critical insight of this study lies in the significantly lower V_S values from experimental measurements compared to theoretical predictions. The reduced V_S profile for MORB, when compared with the 1D PREM model, implies a smaller MORB fraction is needed to explain LLSVP anomalies compared to prior estimates. For instance, Thompson et al. (2019) estimated that -1.5% V_S anomalies would require 64% MORB at 100 GPa or 48% MORB at 125 GPa. I recommend that the authors clarify the distinction between the two uses of $\delta \ln V_S$ and emphasize the comparison of the MORB profile with the 1D average velocity, highlighting the implications of the lower V_S compared with prior results.

3 Choice of CaPv data

While the V_S of the two silica phases reported here is substantially lower than prior calculations, the V_S of MORB is approximately 4% lower than PREM, compared to a 7% reduction for the modeled MORB profile (line 192). This discrepancy suggests that low-velocity phases, particularly Ca-perovskite (CaPv), contribute significantly to the low-velocity MORB profile. Since theoretical and experimental V_s of Capv show significant discrepancies, the required MORB fraction to explain the velocity anomalies strongly depends on the choice of CaPv data. Although Figure S9 shows that different experimental datasets yield minor variations in MORB velocities, the manuscript should explicitly discuss the influence of CaPv data on MORB fraction estimates.

4 Additional comments and suggestions

1. line 26: Since the findings do not exclude thermal anomalies, consider rephrasing this sentence.
2. Line 33: should be *100%
3. Line 99: Reference 49 only calculated velocity changes from stishovite to CaCl₂-type silica.
4. line 138-141: Could you explain why is the transition not observed?
5. line 158: Could you provide an estimate or discussion of uncertainties when extrapolating ambient-temperature data to high temperatures.
6. line 166-199: I recommend splitting this section into two paragraphs—one focusing on the phase transition and another comparing MORB profile with 1D reference models.
7. line 208: It should be -1.5% velocity anomalies instead of discontinuity.
8. line 209-211: It seems these values are from Table S9?
9. line 219: why do the authors choose 3890 K? It seems there is no difference between 3890 and 3900 K.
10. line 236: Include a reference regarding texture development effects.
11. line 268: could you elaborate on the greater heat supply?
12. Tables and Figures:
 - Ensure consistent volume units in Tables S2, S10, and S11 for ease of comparison.
 - Fig 1: add plots for the crystal structure for the two silica phases
 - Fig. 2, 3, S7 and S9: add a depth axis at the top axis
 - There is no orange area in Fig. S9.
 - Table S6-S8 have the same titles and slightly different captions. I recommend removing repeated sentences and highlighting the differences in each table.

'Ultrahigh-Pressure Sound Velocities of Dense SiO₂ Phases and the Origin of the LLSVPs'

by Krymarys et al.

While the authors have made efforts to address most of the points raised during the review, many of the points have not been adequately handled and some inconsistencies still persist. Unfortunately, some of the responses are unnecessarily lengthy and do not contribute to clarify the points because the changes in the actual manuscript text are sometimes minimal. Also, some of the additions in the text in response of the comments are misplaced and seriously affect the flow of the manuscript. Therefore, I cannot recommend the publication of the manuscript in the current state.

Here are some of the points of concern that have not sufficiently or adequately clarified:

- 1) Lines 155-168: The discussion about the interferometer models in the comparison of the present results with those from Asahara's et al (2013) data is very technical and unnecessary as it does not bring much to explain the divergence as the velocities should remain the same regardless of the instrument generation (as acknowledged by the authors...). Therefore, the addition is unnecessary and lines 155 to 164 should be removed to focus the discussion on more realistic explanations for the disagreement.
- 2) Following up the previous point, the sentence in lines 164- ('Therefore, the results of Asahara et al. 2013 may not fully account for the effects of elastic anisotropy, potentially leading to an overestimated dataset compared to our result') is unclear and should be reworked. In fact, I found much more convincing the arguments in the response that the additions in the manuscript related to this point. Among the possible arguments discussed, only potentially the limited sampling in Asahara et al (2013) is a solid argument but it is difficult to read this from the above-mentioned sentence in the text... Also, I still think that the authors should be more conservative in the estimation of errors to account from all plausible sources of uncertainty as a difference of 150 m/s is not unrealistic when dealing with non-annealed polycrystalline materials at the extreme high pressures of this study.
- 3) While I appreciate the efforts to clarify the temperature projection of the velocities in MORB, the text in lines 288-318 is clearly misplaced in the manuscript and cuts the flow of the discussion. It should not be included in the discussion about the origin of the LLSVP but rather integrated in the section related to modelling the V_s profiles of MORB.
- 4) Line 144. – should be 'X-ray diffraction'. However, the reference from Shieh et al 2022 is not appropriate here because the direct data from their XRD study does not show any evidence for softening (i.e., average Q(hkl) increases with increasing pressure, even across the ferro-elastic transition), it is only when the differential stress τ is calculated, which involves G, that there is softening. Therefore, the softening is related to G (which is not obtained in their study) not to the actual parameters measured in the study and hence, the XRD study should not be mentioned here as evidence for softening in experimental studies. In my opinion the paper by Zhang et al. (2021) is already providing the clearest experimental evidence for softening.

There are other additions/responses related to the comments by Reviewer 2 (that I have been asked to check) that are only partially addressed.

- 1) The lengthy response related to the effects of stress and texture in the Brillouin signal does not unfortunately help to clarify this point to potential readers because these arguments are not included in the text (only partially in the Supplementary Materials). I would recommend that authors directly address these concerns in the main text (in a more direct and concise manner), possibly in the section explaining the measurements, as a disclaimer to remove all potential concerns for the rest of the discussion.
- 2) Concerning the pressure determination, I agree with the authors that access to XRD analysis for pressure calibration are not routinely available but they should not claim that there is no standard laboratory-based Brillouin setup inherently equipped with an integrated XRD system, because a system of this type is operated for more than 10 years at BGI-University of Bayreuth. Nevertheless, I would recommend that the authors acknowledge the limitations of the Raman edge approach and the large uncertainties associated to the method particularly in the pressure range below Mbar. Did the authors cross-checked the pressure given by the diamond edge and calculated from the EoX of B2-NaCl in the few XRD data that the authors could collect? This would be a good support to demonstrate that the pressure is properly determined (or at least to have an idea of the deviation).
- 3) Regarding the role of Al, without asking for the experimental results, I think the author could include a comment on this in the discussion about the origin of the LLSVPs, as there is enough information about the effect of Al (changing the transition pressures, decreasing the shear modulus) for a qualitative discussion.

In summary, the authors should still rework the main text to clarify the most important points that seems to be better handled in the responses than in the text. This approach does not contribute to the clarity of the manuscript.

‘Ultra-high-Pressure Sound Velocities of Dense SiO₂ Phases and the Origin of the LLSVPs’
by Krymarys et al.

The manuscript submitted by Krymarys et al. report unique sound velocity measurements on ultrahigh pressure polymorphs of SiO₂, CaCl₂- and α -PbO₂- structured SiO₂) to 150 GPa at room temperature by Brillouin scattering spectroscopy. The results reveal a drop in velocities across the CaCl₂- to PbO₂ structured SiO₂ that further contribute to decrease the velocities of subducted MORB at the bottom of the mantle. Based on this observation, together with modelling of MORB velocities along different thermal regimes (slab vs. mantle geotherm), the authors propose that the Large Low Shear Velocity Provinces (LLVSP) at the bottom of the mantle could be explained by the presence of ~23-33% subducted oceanic crust.

The topic is highly relevant and of potential interest for the readership of COMMSENV. The paper is well crafted and the figures and supported materials are appropriate. However, despite the novelty of the data, the conclusions do not appear supported by the reported data and may rather results from the modelling and associated assumptions. There are also some unexplained inconsistencies between the reported datasets and the previous studies (some of them also ignored here) that have not been addressed in the manuscript as I point out in the detailed comments below. Moreover, the data has not been made available, which difficult the verification of the claims. Based on these concerns, I cannot recommend the acceptance of the manuscript in the present form.

We would like to thank Reviewer#1 for providing us with comments and suggestions for this revision. Below, we address all comments and indicate the corresponding revised lines in the manuscript.

We would also like to emphasize that all experimental data e.g. Fig. 1-2, Fig. S1-S6, S10-S11 and Tables S1-S2 as well as supporting input parameters for BurnMan modelling e.g. MORB composition in Table S3, thermodynamic parameters in Tables S4-S5 and temperature profiles Fig. S8 are included in the manuscript and supplementary materials, which were used for our modeled MORB, including experimentally determined SiO₂ phase, in Fig. 3-5, Fig. S7, S9 and Tables S6-S9.

If Reviewer #1 requires any specific information, we would be happy to provide it upon request.

Below there is a list of the major and minor concerns about the data, methodology and implications:

- 1) The paper does not seem to give proper credit to previous experimental studies of the high-pressure elasticity of SiO₂ polymorphs stishovite and CaCl₂-structured phase. I am referring here to the single-crystal elasticity data of Zhang et al. (PRL 2021) up to 70 GPa, which should be properly acknowledged in the introduction and included in the comparison reported in Fig.2 (see minor comments below). It is not clear why these valuable data have been excluded and the comparison is restricted to previous own data and theoretical calculation. This is not a very scholarly approach.

We apologize for the omission of data from Zhang et al. 2021. In the previous version, our focus was primarily on polycrystalline studies; however, we acknowledge that incorporating the data from Zhang et al. 2021, which focuses on averaged SiO₂ aggregates, provides valuable context for interpreting our results under similar experimental conditions.

Furthermore, the data from Zhang et al. 2021 appear to align well with the previous polycrystalline study by Asahara et al. 2013 and with our study in the ~30–70 GPa pressure

range. Notably, the discrepancy in V_s between Zhang et al. 2021 and Asahara et al. 2013 increases up to ~ 0.15 km/s within the lower pressure regime of ~ 0 –30 GPa.

In the revised manuscript, we have now included the averaged SiO_2 aggregate data from Zhang et al. 2021 in Fig. 2 and have appropriately cited the reference throughout the text.

Revised lines: 90, 144–145, Fig. 2 (line 563).

- 2) The authors emphasize that their results for CaCl_2 -phase reveal velocities that are substantially lower than predicted by theory, which would support the potential role of subducted MORB in explaining the negative velocity anomalies in the LLSVPs. However, they appear to avoid the comparison with previous own data on polycrystalline CaCl_2 - SiO_2 (Ref. 28, Asahara et al. 2013), which are substantially higher than those reported in the present study. The comparison is reduced to the extrapolated velocities, which depend on the fitted shear elastic moduli and pressure derivatives as claimed by the authors (lines 144-148). However, the direct comparison of the velocities in the pressure range of overlap (around 60 GPa) shows differences which are well beyond the uncertainties in shear velocities reported (Suppl. Table S1). How can the authors explain the inconsistencies with previous own measurements where the experimental protocols would be assumed as homogenous? This should be discussed in the manuscript rather than ignored because the disagreement is obvious in Fig.2 and the lack of comparison makes the results suspicious.

In response to this comment, the discussion related to V_s differences in our data and the previously collected data by Asahara et al., 2013, in particular at 60 GPa is now revised and elaborated in lines 155-168 in the manuscript.

Unlike Asahara et al., 2013, who measured only one angular orientation throughout all pressure points, our study included measurements at several angles at each pressure point. Specifically, at ~ 60 GPa (our P1 at 61 GPa, Table S1), we collected 6 measurements at different angles (Table S1) and averaged them (Table S1, Fig. 2), including standard deviation on V_s . Therefore, the results of Asahara et al. 2013 may not fully account for the effects of elastic anisotropy, potentially leading to an overestimated dataset based on our findings.

We highlight that the advanced instrumentation used in this study, including a multi-pass tandem spectrometer with a TFP-2 HC interferometer, provided significant improvements over the previous setup employed by Asahara et al. 2013, which utilized a multi-pass tandem system based on a TFP-1 interferometer. The transition from TFP-1 to TFP-2 HC resulted in a significantly enhanced contrast ratio, reduced background noise, and improved symmetry of the instrumental response function, leading to a higher signal-to-noise ratio in our measurements (Ghost 7.0 Multichannel Analyser – User Guide; Tandem Fabry-Perot Spectrometers TFP-1 and TFP-2 HC – Operator Manual). With this upgrade, data quality improves, but the absolute velocity is unlikely to change.

“The measurements performed at several angular orientations, along with the use of advanced instrumentation in this study - including a multi-pass tandem spectrometer with TFP-2 HC interferometer - provided significant improvements over the previous setup and measurements used by²⁸, which utilized a multi-pass tandem system based on TFP-1 interferometer. The transition from TFP-1 to TFP-2 HC resulted in a significantly enhanced contrast ratio, reduced background noise, and improved symmetry of the instrumental response function, leading to a higher signal-to-noise ratio in our measurements (Ghost 7.0 Multichannel Analyser – User Guide; Tandem Fabry-Perot Spectrometers TFP-1 and TFP-2 HC

– Operator Manual). With this upgrade, data quality improves, but the absolute velocity is unlikely to change. Therefore, the results of Asahara et al. 2013²⁸ may not fully account for the effects of elastic anisotropy, potentially leading to an overestimated dataset compared to our result. These approaches collectively may further explain the observed differences between our results and those of Asahara et al. 2013²⁸, particularly at ~60 GPa, where the discrepancy reaches up to 0.15 km/s”.

In addition to this, we provide explanation on background subtraction procedure from our obtained Brillouin spectra in the revised manuscript in lines 125-128.

“All acquired spectra were subject to background subtraction, which resulted in a consistent VS reduction within 0.02 km/s in α -PbO₂-type in all collected data, and within a maximum of 0.2 km/s in the CaCl₂-type”.

Specifically, at ~60 GPa (our P1 at 61 GPa, Table S1) the background subtraction resulted in lowering raw Brillouin Vs spectra by 0.084 km/s.

- 3) Following in the previous comment, the only possibilities to explain the disagreement is the quality of the polycrystalline SiO₂ samples (compaction degree, textures, etc) employed here and in Asahara et al. (2013), or by the fact that the present data may suffer from vignetting (Sinogekin et al., 1998), which will result in systematically lower velocities. Therefore, until a reasonable explanation for the inconsistencies in own measurements is not provided, the data reported here is difficult to trust. At least in single-crystal studies there are several examples of interlaboratory consistency in the sound velocity data, and this should be also expected for polycrystalline samples if the experiments are properly conducted.

In response to this comment, we have revised and expanded the discussion on the differences in Vs between our data and the previously collected data by Asahara et al. 2013 in lines 125-128 and 155-168 of the manuscript, as well as in the previous point (2).

Below, we outline key aspects that may have contributed to the observed differences in Vs profiles between our study and that of Asahara et al. 2013:

1. Conducting measurements at multiple angular orientations in this study
2. Adoption of a multi-pass tandem spectrometer with a TFP-2 HC interferometer
3. Background subtraction of acquired Brillouin spectra

- 4) Looking at the velocity data for CaCl₂ and α -PbO₂ at the bottom of the mantle (Fig.2), the velocity contrast is almost unresolvable within mutual uncertainties so the authors should not be putting that much weight on the contrast. Also, it is not clear in the text (lines 133-134) in the reported contrast is derived from the present study or from the cited references. Please clarify. Given the unresolvable contrast, it is not clear to me how this can result in an enhanced contrast along the slab and mantle geotherms as shown in Fig.3, unless this results from the parameters and assumptions adopted in the projection of velocities along the geotherm. This should be clarified.

In response to the mentioned contrast, we have revised the wording to moderate any potential implications. To ensure a more cautious interpretation, we have adopted a speculative tone using terms such as may, might, can, and could in the revised manuscript (lines 134-138, 197, 209, 210, 219, 223, 224, 236, 238, 240, 247, and 256).

The ~2.5-3.6% V_s reduction corresponds to the expected SiO_2 phase transition in the SiO_2 system at 120-125 GPa (Tables S6-S7). This range includes values obtained in this study (Table S6) as well as those from other relevant case studies (Table S7). The revised manuscript clarifies this aspect and specifies that the V_s reduction observed exclusively in this study is 3-3.2%, following cold slab- or lower mantle geotherms (Fig. 4 and Table S6), as noted in the revised line 137.

Additionally, the composition of MORB, the thermodynamic parameters used, and the adopted temperature profiles used for modeling are detailed in the supplement (Tables S3–S4, Fig. S8). This clarification is provided in lines 134–138, directing the reader to the revised lines 179–180, where thermodynamic parameters, shear wave velocity data from this work, temperature profiles, and MORB compositions are referenced alongside their corresponding figures and tables.

Minor comments:

Line 90: Reference Zhang et al. 2021 (*Physical Review Letters*, 126, 025701), which has the added value of being single-crystal elasticity data up to 70 GPa, should be cited here.

The reference to Zhang et al. 2021 has been included in the revised manuscript at lines 90, 144–145, and in Fig. 2.

Line 114-116: Sentence should be reconsidered for clarity: ‘The CaCl_2 -type SiO_2 phase was synthesized in a DAC upon compression above 55 GPa using stishovite, which was pre-synthesized using a large volume press, as starting material (or similar).

The sentence in lines 114-115 has been revised to: “The CaCl_2 -type SiO_2 phase was synthesized in a DAC by compressing stishovite—pre-synthesized in a large-volume press—above 55 GPa”.

Line 139-140: Softening has been also confirmed by single-crystal Brillouin scattering. Ref. Zhang et al. PRL 2021 should be included in the discussion here.

The reference to Zhang et al., 2021 has been included in the revised manuscript at lines 144-145. “This is anticipated from theory to induce a sharp shear softening^{24,50}, and was also shown in an X-ray study of (Shieh et al., 2002)⁵⁷ to occur at 48 GPa, or in a Brillouin study of a single-crystal stishovite at 55 GPa by Zhang et al., 2021⁴⁴ or polycrystalline stishovite from Asahara et al., 2013²⁸ at the pressure range of 25-35 GPa”.

Line 198: should read: ‘...under extreme pressure conditions should...’

This sentence has been revised in line 225.

“Given these results, the newly acquired elasticity data of SiO_2 high-pressure phases under extreme pressure conditions can offer vital insights into a more comprehensive understanding of the seismic structure of LLSVPs”.

Figure 2: The aggregate shear wave velocity data from the single-crystal study of Zhang et al. (PRL 2021) should be included in the comparison. All necessary data for the comparison is available with the manuscript. Also, the lack of consistency with previous experimental studies (Ref. 28) should not be ignored.

Data from Zhang et al. 2021 has now been incorporated into the revised manuscript and Fig. 2. The differences are further elaborated in the revised manuscript, as detailed in points 2-3.

In summary, the authors should provide compelling arguments to ensure the quality of the data reported here and to explain the differences with previous (own) studies. And the fact that velocity contrast appears resolvable when projected along the geotherms when it is not a clear feature in the original room temperature data.

To address the room-temperature effect on our modeled MORB, we have added an additional figure (Fig. 3) presenting the results at room temperature. As a result, all subsequent figures in the revised manuscript have been renumbered. Additional clarification has also been included in lines 183, 198-200.

“We found that the phase transition from CaCl₂-type to α -PbO₂-type in the MORB and pure SiO₂ system could lead to discontinuous shear wave velocity reductions, reaching a maximum of 0.6% and 3.2%, respectively (Fig. 4, Table S6). If we assume the modeled MORB under ambient temperature (Fig. 3), the contribution of SiO₂ phase transition, as the negative discontinuity feature in MORB, would decrease to ~0.33%.

In this manuscript, Krymarys et al. present new experimental measurements of the shear-wave velocity (VS) of two high-pressure silica phases at pressures up to the core-mantle boundary (CMB). The reported VS values for these silica phases are significantly lower (by approximately 7–14%) than previous theoretical predictions from first-principles calculations. Utilizing these measurements, the authors model the VS profile of mid-ocean ridge basalt (MORB) assuming varying silica contents. Their results indicate a velocity reduction of 0.6% or 1.1–1.3% due to the phase transition from CaCl-type to α -PbO-type silica. Furthermore, they find that the VS of MORB, following a cold slab geotherm, is approximately 7% lower than the PREM model. Consequently, they estimate that a MORB fraction of ~23–33% is required to explain the observed -1.5% low-velocity anomalies within LLSVPs. Based on these results, the authors propose that ancient subducted MORB, even without significant thermal anomalies, may account for LLSVP characteristics.

The methodological approach and workflow are clear, and the new findings contribute significantly to the ongoing debate regarding the origin of LLSVPs. Nonetheless, I recommend acceptance conditional upon major revisions addressing the following substantive issues to enhance the clarity, precision, and impact of this contribution.

We would like to thank Reviewer#3 for providing us with comments and suggestions in this revision. Below, we address all comments and indicate the corresponding revised lines in the manuscript.

1 Misuse of $\delta \ln VS$

The manuscript uses $\delta \ln VS$ to describe both the velocity anomalies within LLSVPs (e.g., lines 33, 212) and the negative velocity change (e.g., 177) associated with phase transitions. This dual usage may confuse readers. For example, line 208 incorrectly refers to LLSVP velocity anomalies as a “discontinuity”. The -1.5% $\delta \ln VS$ represents deviations relative to an average reference velocity at the same depth, whereas a discontinuity denotes a velocity change across a phase transition depth. The authors should clearly distinguish these concepts throughout the text.

In response to this comment, the concept of discontinuity has now been introduced in lines 135-136.

To ensure clarity and consistency, all our data, supported by modeling, now explicitly refer to the discontinuity, where SiO₂ may contribute to the observed Vs reduction across the transition from the CaCl₂-type to the α -PbO₂-type phase.

References to the discontinuity and the SiO₂ phase transition have been carefully incorporated throughout the manuscript in lines 135, 197, 200, 239, 247, 257, 267, and Fig. 5 (lines 601-603).

2 Focus on the comparison of MORB profile with 1D reference model

The velocity reduction caused by the phase transition is minor, approximately 1% or less in MORB, and decreases further when considering MORB fractions within the mantle. Consequently, the phase transition’s effect on the overall velocity profile is negligible. In my opinion, the critical insight of this study lies in the significantly lower VS values from experimental measurements compared to theoretical predictions. The reduced VS profile for MORB, when compared with the 1D PREM model, implies a smaller MORB fraction is needed to explain LLSVP anomalies compared to prior estimates. For instance, Thompson et al. (2019) estimated that -1.5% VS anomalies would require 64% MORB at 100 GPa or 48% MORB at 125 GPa. I recommend that the authors clarify the distinction between the two uses of $\delta \ln$

VS and emphasize the comparison of the MORB profile with the 1D average velocity, highlighting the implications of the lower VS compared with prior results.

We agree with the Reviewer#3 that the comparison of our experimentally obtained data on SiO₂ phases under ambient temperature provides a valuable opportunity to directly compare with theoretical studies and assess variations that have not been previously available.

Theoretical G_0 and G_0' values tend to be overestimated compared to experimental values, such as in case of CaPv e.g. Greaux et al., 2019, Thomson et al., 2019 or our experimentally determined SiO₂ phases in this study. Consequently, using these theoretical parameters in modeled MORB compositions, particularly for dominant phases, can lead to an elevated V_S profile and an overestimated MORB volume fraction required to explain the observed -1.5% anomaly. Additionally, discrepancies between studies, such as our work and those by Thomson et al. 2019 and Wang et al. 2020, may arise from differences in the partitioning behavior and proportions of mineral phases used in MORB modeling.

In this study, we investigate the implications of the SiO₂ phase transition using experimentally determined mineral phases of MORB, incorporating updated constraints on G_0 and G_0' for all bridgmanite endmembers Murakami et al., 2024. The lack of high-temperature experimental data on MORB's mineral phases under lower mantle conditions presents challenges in reconciling discrepancies with previous studies. While Mattern et al. 2005 emphasized the primary influence of pressure on the V_S profile, the role of G_0' remains critical for interpreting seismic observations. Furthermore, previous studies e.g., Stixrude et al., 2011; Murakami et al., 2012 suggest that temperature sensitivity in η_{50} can introduce experimental uncertainties of ~10%. In our study, such uncertainties correspond to a ± 1 vol.% variation in MORB content needed to explain the observed -(1.5–3)% discontinuities.

If the CaPv content in MORB increases by 7 wt.% (from 23 wt.% in Table S3 to 30 wt.%), as suggested by Ricolleau et al. 2010, while the SiO₂ phase decreases from 17 wt.% to 10 wt.% - a value significantly lower than those reported in other studies e.g., Hirose et al., 2005; Perrillat et al., 2006; Ricolleau et al., 2010; Ishii et al., 2022 - then the same MORB volume fraction could still account for the discussed discontinuities. This adjustment would slightly reduce the SiO₂ contribution from ~0.6% (Fig. 4, Table S6) to ~0.5%.

These results highlight the importance of experimental constraints in refining theoretical models and interpreting seismic profiles in the lower mantle. Although the effect of the SiO₂ phase transition (CaCl₂-type to α -PbO₂-type) is relatively small (~0.6 - 1%, Tables S6 - S8) in modeled MORB compositions under assumed cold slab or lower mantle geotherms - and further diminishes to ~0.33% under assumed ambient conditions (Fig. 3) - the collective contribution of MORB phases with experimentally refined G_0' values provide valuable insight into how a realistic V_S profile of MORB can decrease relative to the PREM model. This study contributes to understanding the potential role of MORB in generating negative anomalies in the range of -(1.5–3)%.

A detailed discussion of these aspects is provided in the Origin of LLSVPs section (lines 288-318):

References

Gréaux, S., Irifune, T., Higo, Y., Tange, Y., Arimoto, T., Liu, Z., & Yamada, A. (2019). Sound velocity of CaSiO₃ perovskite suggests the presence of basaltic crust in the Earth's lower mantle. *Nature*,

- 565(7738), 218–221. <https://doi.org/10.1038/s41586-018-0816-5>
- Thomson, A. R., Crichton, W. A., Brodholt, J. P., Wood, I. G., Siersch, N. C., Muir, J. M. R., et al. (2019). Seismic velocities of CaSiO₃ perovskite can explain LLSVPs in Earth's lower mantle. *Nature*, 572(7771), 643–647. <https://doi.org/10.1038/s41586-019-1483-x>
- Hirose, K., Takafuji, N., Sata, N., & Ohishi, Y. (2005). Phase transition and density of subducted MORB crust in the lower mantle. *Earth and Planetary Science Letters*, 237(1–2), 239–251. <https://doi.org/10.1016/j.epsl.2005.06.035>
- Ishii, T., Miyajima, N., Criniti, G., Hu, Q., Glazyrin, K., & Katsura, T. (2022). High pressure-temperature phase relations of basaltic crust up to mid-mantle conditions. *Earth and Planetary Science Letters*, 584, 117472. <https://doi.org/10.1016/j.epsl.2022.117472>
- Mattern, E., Matas, J., Ricard, Y., & Bass, J. (2005). Lower mantle composition and temperature from mineral physics and thermodynamic modelling. *Geophysical Journal International*, 160(3), 973–990. <https://doi.org/10.1111/j.1365-246X.2004.02549.x>
- Murakami, M., Ohishi, Y., Hirao, N., & Hirose, K. (2012). A perovskitic lower mantle inferred from high-pressure, high-temperature sound velocity data. *Nature*, 485(7396), 90–94. <https://doi.org/10.1038/nature11004>
- Murakami, M., Khan, A., Sossi, P. A., Ballmer, M. D., & Saha, P. (2024). The Composition of Earth's Lower Mantle. *Annual Review of Earth and Planetary Sciences*, 52(1), 605–638. <https://doi.org/10.1146/annurev-earth-031621-075657>
- Perrillat, J. P., Ricolleau, A., Daniel, I., Fiquet, G., Mezouar, M., Guignot, N., & Cardon, H. (2006). Phase transformations of subducted basaltic crust in the upmost lower mantle. *Physics of the Earth and Planetary Interiors*, 157(1–2), 139–149. <https://doi.org/10.1016/j.pepi.2006.04.001>
- Ricolleau, Perrillat, J. P., Fiquet, G., Daniel, I., Matas, J., Addad, A., et al. (2010). Phase relations and equation of state of a natural MORB: Implications for the density profile of subducted oceanic crust in the Earth's lower mantle. *Journal of Geophysical Research: Solid Earth*, 115(B08202). <https://doi.org/10.1029/2009JB006709>
- Stixrude, L., & Lithgow-Bertelloni, C. (2011). Thermodynamics of mantle minerals - II. Phase equilibria. *Geophysical Journal International*, 184(3), 1180–1213. <https://doi.org/10.1111/j.1365-246X.2010.04890.x>
- Thomson, A. R., Crichton, W. A., Brodholt, J. P., Wood, I. G., Siersch, N. C., Muir, J. M. R., et al. (2019). Seismic velocities of CaSiO₃ perovskite can explain LLSVPs in Earth's lower mantle. *Nature*, 572(7771), 643–647. <https://doi.org/10.1038/s41586-019-1483-x>
- Wang, Xu, Y., Sun, D., Ni, S., Wentzcovitch, R., & Wu, Z. (2020). Velocity and density characteristics of subducted oceanic crust and the origin of lower-mantle heterogeneities. *Nature Communications*, 11(1), 1–8. <https://doi.org/10.1038/s41467-019-13720-2>

3 Choice of CaPv data

While the VS of the two silica phases reported here is substantially lower than prior calculations, the VS of MORB is approximately 4% lower than PREM, compared to a 7% reduction for the modeled MORB profile (line 192). This discrepancy suggests that low-velocity phases, particularly Ca-perovskite (CaPv), contribute significantly to the low-velocity MORB profile. Since theoretical and experimental Vs of Capv show significant discrepancies, the required MORB fraction to explain the velocity anomalies strongly depends on the choice of CaPv data. Although Figure S9 shows that different experimental datasets yield minor variations in MORB velocities, the manuscript should explicitly discuss the influence of CaPv data on MORB fraction estimates.

Indeed, the V_s reduction in MORB is nearly twice that of pure SiO₂ when compared to PREM, assuming both follow the same temperature profile.

CaPv contributes to the overall V_s reduction in MORB, assuming that the recently determined G_0 and G_0' values (Greaux et al., 2019; Thomson et al., 2019) - derived from limited experimental data and extrapolation - accurately represent CaPv behavior in the lower mantle.

In our modeling, we adopt the recently reported lower G_0 and G_0' values for cubic CaPv (Greaux et al., 2019) (see Table S6) and assume a 23 wt.% CaPv proportion, based on experimental MORB data from Hirose et al., 2005 at 60 GPa. This is consistent with Ishii et al. 2022, who reported a similar CaPv proportion of 22 wt.% in MORB.

Thomson et al. 2019 considered a simplified model with a higher CaPv proportion of ~30 vol.% but did not account for endmember partitioning, particularly for bridgmanite, the dominant mineral phase. Their choice of G_0' for bridgmanite endmembers may have influenced the MORB V_s profile and, consequently, the estimated MORB volume fraction needed to explain the observed -1.5% anomaly. This could explain the differences between our study and Thomson et al., 2019.

Other experimental studies on MORB, such as those by Ricolleau et al. 2010 and Perrillat et al. 2006, reported higher CaPv proportions (31 wt.% and 27 wt.%, respectively) in bridgmanite-enriched and CF-depleted compositions. These compositions would require a greater MORB volume fraction (~40 vol.% under lower mantle conditions and ~50 vol.% under a cold slab geotherm) to explain the observed -1.5% V_s anomaly. However, the contribution of the SiO₂ phase transition (from CaCl₂-type to α -PbO₂-type) would remain stable, reducing V_s at ~0.6% across these studies. This is consistent with the SiO₂ effect observed in our study, including MORB composition from Hirose et al. 2005 (Table S6).

SiO₂ phase transition contribution to MORB V_s reduction:

1) This study (MORB from Hirose et al., 2005):

0.58% under cold slab geotherm

0.63% under lower mantle geotherm

2) Ricolleau et al., 2010:

0.53% under cold slab geotherm

0.57% under lower mantle geotherm

3) Perrillat et al., 2006 (with endmember partitioning based on Ricolleau et al., 2010):

0.55% under cold slab geotherm

0.60% under lower mantle geotherm

To estimate the influence of CaPv, we have incorporated an analysis in this revision examining the variations of SiO₂ and CaPv in MORB composition from Hirose et al. 2005. This assessment explores how increasing or decreasing the proportion of one mineral phase affects the other.

This estimate is now included in the revised manuscript (lines 304–309).

For this purpose, we considered two cases with ± 6 –7 wt.% variations:

Case 1: Increasing CaPv from 23 wt.% to 30 wt.% while reducing SiO₂ from 17 wt.% to 10 wt.% results in a slight decrease in the SiO₂ phase contribution by 0.1% (from ~0.6% down to ~0.5% under both cold slab and lower mantle geotherms).

Case 2: Decreasing CaPv from 23 wt.% to 17 wt.% while increasing SiO₂ from 17 wt.% to 23 wt.% (consistent with reported SiO₂ enrichment at 113 GPa in Hirose et al., 2005) results in

an increase in the SiO₂ phase contribution by 0.2% (from ~0.6% up to ~0.8% under both geotherms).

Importantly, neither case affects the required MORB volume fraction needed to explain the -1.5% or -3% seismic discontinuities.

Since no experimental studies on MORB (e.g., Perrillat et al., 2006; Ricolleau et al., 2010; Ishii et al., 2022; Hirose et al., 2005) have reported SiO₂ content below 15 wt.%, we also examined an extreme scenario in which SiO₂ is reduced to 10 wt.% to assess the effect of its diminished presence. This was balanced by increasing CaPv to 30 wt.% (consistent with the high CaPv proportion reported by Ricolleau et al., 2010). Even in this scenario, the SiO₂ contribution is only reduced by 0.1% (from ~0.6% to ~0.5%), which remains consistent with the modeled conclusions of this study (see e.g. Table S6).

References

- Gréaux, S., Irifune, T., Higo, Y., Tange, Y., Arimoto, T., Liu, Z., & Yamada, A. (2019). Sound velocity of CaSiO₃ perovskite suggests the presence of basaltic crust in the Earth's lower mantle. *Nature*, 565(7738), 218–221. <https://doi.org/10.1038/s41586-018-0816-5>
- Hirose, K., Takafuji, N., Sata, N., & Ohishi, Y. (2005). Phase transition and density of subducted MORB crust in the lower mantle. *Earth and Planetary Science Letters*, 237(1–2), 239–251. <https://doi.org/10.1016/j.epsl.2005.06.035>
- Perrillat, J. P., Ricolleau, A., Daniel, I., Fiquet, G., Mezouar, M., Guignot, N., & Cardon, H. (2006). Phase transformations of subducted basaltic crust in the upmost lower mantle. *Physics of the Earth and Planetary Interiors*, 157(1–2), 139–149. <https://doi.org/10.1016/j.pepi.2006.04.001>
- Ricolleau, Perrillat, J. P., Fiquet, G., Daniel, I., Matas, J., Addad, A., et al. (2010). Phase relations and equation of state of a natural MORB: Implications for the density profile of subducted oceanic crust in the Earth's lower mantle. *Journal of Geophysical Research: Solid Earth*, 115(B08202). <https://doi.org/10.1029/2009JB006709>
- Thomson, A. R., Crichton, W. A., Brodholt, J. P., Wood, I. G., Siersch, N. C., Muir, J. M. R., et al. (2019). Seismic velocities of CaSiO₃ perovskite can explain LLSVPs in Earth's lower mantle. *Nature*, 572(7771), 643–647. <https://doi.org/10.1038/s41586-019-1483-x>

4 Additional comments and suggestions

1. line 26: Since the findings do not exclude thermal anomalies, consider rephrasing this sentence.

Rephrased to "...without abrupt thermal anomalies (+1500 K),...", lines: 26, 601-603.

2. Line 33: should be *100%

Line 33 is revised.

3. Line 99: Reference 49 only calculated velocity changes from stishovite to CaCl₂-type silica.

Line 99 has been revised, and now only the reference to Yang et al. 2014, currently numbered as reference 50, is included.

4. line 138-141: Could you explain why is the transition not observed?

This aspect is now clarified in the revised manuscript, in lines 146-149.

“Starting this study at a relatively high pressure of 55 GPa in polycrystalline SiO₂ (CaCl₂-type), similar to the material examined by Asahara et al., 2013, may explain why the ferroelastic transition - previously observed at 25–35 GPa - was not detected in our work”.

5. line 158: Could you provide an estimate or discussion of uncertainties when extrapolating ambient-temperature data to high temperatures.

Yes, this aspect has been incorporated into the revised manuscript in lines 134-138 and 198-200.

Figure 2 illustrates a ~1.5% reduction in V_s across the SiO₂ phase transition within the 120-125 GPa range. Under modeled cold slab or lower mantle geotherms, this reduction increases to approximately 3-3.2% in a pure SiO₂ system (Tables S6, S7).

For MORB modeling:

Under ambient temperature conditions, the V_s reduction across the SiO₂ phase transition is ~0.33% (newly added Figure 3).

Under cold slab or lower mantle geotherms (previously Figure 3, now Figure 4), the V_s reduction increases to ~0.6%.

6. line 166-199: I recommend splitting this section into two paragraphs—one focusing on the phase transition and another comparing MORB profile with 1D reference models.

The section is revised, line 217.

The comparison with PREM is now presented in a separate paragraph titled 'Comparison with PREM', which follows the previously combined and extensive paragraph 'Modeling of V_s Profile of MORB'.

7. line 208: It should be -1.5% velocity anomalies instead of discontinuity.

This is revised, currently in line 235.

8. line 209-211: It seems these values are from Table S9?

The estimate of 23–33 vol.% MORB is based on the range of MORB compositions considered in Hirose et al. 2005. One boundary is defined by bridgmanite endmembers from the theoretical study of Stixrude et al. 2011, while the other boundary is derived from the same MORB composition but incorporates bridgmanite endmembers from recent experimental studies.

Table S9 compiles the various examined cases, confirming that these values are documented there. Accordingly, the revised manuscript (lines 237–243) now includes references to Tables S6–S9.

9. line 219: why do the authors choose 3890 K? It seems there is no difference between 3890 and 3900 K.

Indeed, the difference between 3890K and 3900K is negligible. The revised manuscript now reflects this by including the temperature range of 3890-3900K, as noted in line 246.

10. line 236: Include a reference regarding texture development effects.

Texture development is now revised and supported with studies from Oganov et al., 2005 and Walte et al., 2009. These references are now added to the revised version of manuscript, current line 263.

References

- Oganov et al., Martoňák, R., Laio, A., Raiteri, P., & Parrinello, M. (2005). Anisotropy of earth's D'' layer and stacking faults in the MgSiO₃ post-perovskite phase. *Nature*, 438(7071), 1142–1144. <https://doi.org/10.1038/nature04439>
- Walte et al., Heidelbach, F., Miyajima, N., Frost, D. J., Rubie, D. C., & Dobson, D. P. (2009). Transformation textures in post-perovskite: Understanding mantle flow in the D'' layer of the Earth. *Geophysical Research Letters*, 36(4), 3–7. <https://doi.org/10.1029/2008GL036840>

11. line 268: could you elaborate on the greater heat supply?

Lines 327-328 in the revised manuscript now provide additional clarification and elaboration on this aspect.

A key finding of our study is that the new elastic wave velocity results for SiO₂ allow us to explain LLSVPs without requiring an unrealistic +1500 K temperature increase at the LLSVP boundary. While we do not rule out the possibility of temperature anomalies, our findings significantly reduce the necessity of invoking previously debated extreme and abrupt thermal boundary conditions, particularly in explaining the -1.5% velocity discontinuity observed in this study.

This is reflected in the revised text:

'This may suggest the presence of enhanced thermal flux in certain regions, although without inducing abrupt thermal anomalies of +1500 K, and could potentially lead to significant mantle upwelling flows.

12. Tables and Figures:

Ensure consistent volume units in Tables S2, S10, and S11 for ease of comparison.

Table S2 has been revised to reflect α -PbO₂-type in cm³/vol, along with the corresponding update in Fig. S11.

Fig 1: add plots for the crystal structure for the two silica phases

Figure 1 has been revised to incorporate the suggested addition of crystal structures for the two silica phases.

Fig. 1. Brillouin spectra of high-pressure SiO_2 phases loaded with NaCl B2 as pressure medium in the lower mantle. a, b, polycrystalline CaCl_2 -type SiO_2 phase at 57 and 126 GPa and 300 K; c, d, polycrystalline $\alpha\text{-PbO}_2$ -type SiO_2 phase at 43 and 148 GPa and 300 K. Atomic distributions in the orthorhombic lattices of CaCl_2 -type (upper, green) and $\alpha\text{-PbO}_2$ -type (lower, red) structures, where dark green/grey atoms represent silicon and green/red atoms represent oxygen (*from Vesta*).

Fig. 2, 3, S7 and S9: add a depth axis at the top axis

Since depth and pressure are not linearly correlated, our primary focus in this modeling was to determine the V_s of MORB under ambient, cold slab, and lower mantle temperature conditions (Fig. 3-4) as a function of pressure.

For ambient temperature conditions in pure SiO_2 samples (Fig. 2), if a linear correlation between pressure and depth is assumed, estimated depths could be included. In this rebuttle, we have tentatively applied the PREM relationship to provide an estimated depth-pressure correlation to Fig. 2.

However, Figs. 4, S7, and S9 present more complex cases, as they examine SiO_2 and MORB systems under cold slab and lower mantle geotherms. Fig. 3 also represents MORB systems, along SiO_2 panels, under ambient conditions. Accurately representing depth correlations for MORB compositions under these temperature profiles would require a significant increase in the number of figures, essentially doubling them (e.g., Fig. 3, Fig. 4 (formerly Fig. 3), as well

as Figs. S7 and S9) to separately illustrate V_s dependence on depth under different thermal conditions.

We have instead provided panel b in Fig. S8 presenting the input data for the examined temperature profiles by visualizing the pressure-depth relationship under these geotherms.

Fig. 2. Shear wave velocity profiles of high-pressure SiO_2 phases under high-pressures. The bold lines are the fitted curves by the third-order finite strain equation for the experimental study⁴⁸. Dotted lines represent the shear wave velocity profiles from the computational studies^{24,50}. Dashed line represents the fitted curve of shear wave velocity profile from the experimental study of²⁸. The grey shaded area indicates the possible phase transition pressure range from CaCl_2 -type to $\alpha\text{-PbO}_2$ -type SiO_2 estimated from previous experiments^{45–48}. The blue shaded area indicates the expected pressure range where the LLSVPs are primarily observed⁵.

Fig. S8. The adopted temperature profiles in this study: cold slab⁴⁰ and lower mantle²⁷ geotherms a), and the correlation between pressure and depth in the examined temperature profiles b). The geotherms in this study are compared to Brown and Shankland profile⁴⁴, adopted as a normal geotherm in theoretical study of⁴⁵, and to Brown and Shankland profile⁴⁴, reduced by 500 K, and adopted as a cold geotherm in theoretical study of⁴⁵.

There is no orange area in Fig. S9.

Fig. S9 has been revised to show the orange area.

Table S6-S8 have the same titles and slightly different captions. I recommend removing repeated sentences and highlighting the differences in each table.

In response to the Reviewer's suggestions, Tables S6-S8 in the revised supplement have been adjusted and simplified accordingly.

Reviewer #2 (Remarks to the Author):

The authors present Brillouin light scattering results of polycrystalline SiO₂ phases at high pressure. The Brillouin results are used to derive V_s of the SiO₂ phases and modelled thermo-elastically at high P-T. The results are applied to understand large low shear velocity provinces in the deeper lower mantle. There are a number of technical and scientific issues that would prevent the work from being published.

These issues are discussed in more details below.

Firstly, polycrystalline Brillouin data does not truly represent aggregate V_s. Brillouin spectral intensity is a function of crystallographic orientation and thus is an assemble of all scattered signals from the compressed sample. In this study, no pressure medium was used even though the authors claim that NaCl is used, but it's not a good medium unless it's annealed. Stress and textures can be developed in this type of samples that would further complicate the issue. The interpretation of the spectra is thus a complex issue. Just look at widths and shapes of the spectra. There are some limited studies in the field on this topic, but this issue is not settled as it is sample dependent.

We appreciate Reviewer #2's comment regarding the interpretation of Brillouin spectra from polycrystalline samples and the potential effects of stress and texture. While it is true that Brillouin intensity depends on crystallographic orientation, the approach of measuring polycrystalline samples has been widely used and validated in high-pressure elasticity studies (e.g., Sinogeikin et al., 2004, Asahara et al., 2013, Buchen et al., 2018, Murakami et al., 2012, Dai et al., 2013, Dubrovinsky et al., 2001).

Brillouin scattering measurements on polycrystalline samples are commonly used to estimate aggregate velocities. The assumption is that the measured spectra capture a statistically representative set of grain orientations, which allows for a reasonable estimate of the effective aggregate velocities (e.g. Sinogeikin et al., 2004).

While single-crystal measurements provide full elasticity tensors (C_{ij}), polycrystalline Brillouin data are still valuable for determining aggregate properties.

NaCl was used, as seen in Fig. 1a,c and Figs. S3-S4. Although detected with a relatively weak signal due to its thin thickness, it was added upon loading to avoid obscuring the signal from the main sample. Its presence indicates that the pressure medium was successfully loaded into our samples.

NaCl is a solid pressure medium widely adopted in high-pressure mineral physics (e.g. Asahara et al., 2013, Andrault et al., 2003). It was chosen to ensure the stability of the NaCl B2 phase throughout all experiments conducted above 35 GPa (e.g. Murakami et al., 2019).

Laser annealing is known to reduce stress in diamond anvil cell (DAC) experiments, often halving or even reducing it to one-third of initial values (Tateno et al., 2019), indirectly implying that laser annealing enhances hydrostaticity in the measured samples. For SiO₂ phases, laser annealing has been observed to lower deviatoric stress (Andrault et al., 2003; Asahara et al., 2013; Tateno et al., 2019), though the effects above 60 GPa remain poorly quantified due to limited reference points and insufficient annealing time (Andrault et al., 2003; Asahara et al., 2013). While stress is reported to improve hydrostaticity by (Andrault et al., 2003) with the use of NaCl as a pressure medium, its effect is not quantified. Moreover, stress values reported in (Asahara et al., 2013) with the use of NaCl as a pressure medium at 39 and 64 GPa – 4.4 GPa and 7 GPa, respectively – are consistent with non-hydrostatic stress values of 5(±2) and 8(±4) GPa at 40 GPa and 60 GPa, as reported by (Shieh et al., 2002).

Indeed, the annealing could have improved the obtained deviatoric stress (t) values obtained at our high-pressure conditions of 92 and 99 GPa in CaCl₂-type and α -PbO₂-type, with values of $t=5$ GPa and $t=6$ GPa, respectively (Supporting Information, 5) Estimation of Stress Conditions). However, we provide several ways in which we support that the stress did not develop with increasing pressure such as the correlation between the hydrostaticity within the sample and the pressure distribution/gradient over the sample (e.g. Boehler et al., 2000, Uts et al., 2013, Smith et al., 2018). This implies that the non-hydrostaticity can be considered to be well suppressed if the pressure gradient over the DAC sample is reasonably small. The pressure distribution within the sample area (~ 20 μm in size) where we performed all the measurements including Brillouin, Raman, and X-ray falls within the range of 1-2 GPa variation within a single pressure point consisting of several angles (Table S1, Supplementary Text, 2), which is reasonably small. This observation would be also indirect supportive evidence that the non-hydrostaticity in our DAC samples remained well suppressed.

In addition to this, the differential stress in CaCl₂-type sample, not in the pressure medium, was assessed by following the methodology outlined by (Shieh et al., 2002). The differential stress in CaCl₂-type sample can be estimated from the shear modulus and the average Q hkl value from all measured reflections, following equation:

$$t = 6G\{Q(hkl)\}$$

The estimated stress calculation within the sample of CaCl₂-type from the pressure condition measured at synchrotron at 92 GPa equals to $t = 2.03$ GPa, which is slightly below the value of estimated stress condition in NaCl-B2 pressure medium ($t \sim 5$ GPa). This agrees well with the consensus on pressure medium effect that fully surrounds sample in the central location, where stress condition is equivalent to the one in pressure medium or it is slightly lower (Supporting Information, 5) Estimation of Stress Conditions).

In addition to this, we examined how a pressure change of ~ 5 GPa and ~ 6 GPa at pressure of ~ 100 GPa affects V_s using the obtained shear wave velocity profile from finite strain fitting. As a result, we found that, under pressure at 100 GPa, the effect on V_s in CaCl₂ sample is found to be 0.46% (± 0.035 km/s in V_s), while the effect on V_s in α -PbO₂ sample 0.68% (± 0.05 km/s), which are below the experimental errors of V_s we determined (Table S1), (Supporting Information, 5) Estimation of Stress Conditions).

The collected Brillouin data consist of several, up to 6 angular orientations, in both CaCl₂-type and α -PbO₂-type samples (Table S1). The collected data show sharp TA modes of the measured samples (Fig. 1). To avoid any unwanted crystallographic orientation, having measured several angles, allowed us to provide an averaged V_s values for both samples at each measured pressure condition (Table S1). As mentioned above, there was no systematic increase in stress or peak broadening as a function of pressure.

While we recognize the complexities involved in interpreting polycrystalline Brillouin data, particularly in challenging materials under ultrahigh-pressure conditions, we have adhered to established methodologies in mineral physics to ensure the reliability of our results. So far, no V_s data are available for polycrystalline SiO₂ above 65 GPa. To improve data accuracy and interpretability, we extended the measurement duration over several days for each angular orientation and measured multiple orientations at each examined pressure condition. This approach allowed us to collect high-quality spectra with well-defined peaks, ensuring that our analysis is based on robust and representative data.

Secondly, pressure was not determined reliably. The authors used Raman edge which is mean to be a secondary option (when there's no other option available for P determination). Diamond Raman edge can be influenced by stress environment and diamond itself, and has lots of uncertainties to it.

There's a precision vs accuracy issue in what exactly the pressure and its uncertainty the authors are reporting. The authors could have used XRD of their samples for P determination, but they did not.

We appreciate Reviewer #2's concerns regarding the use of the diamond Raman T_{2g} mode for pressure determination. While we acknowledge that non-hydrostatic stress conditions can introduce uncertainties in Raman-derived pressures, this method remains widely used in high-pressure studies, particularly when X-ray diffraction (XRD) measurements are not feasible at every pressure step. Although some specialized synchrotron-based setups allow for simultaneous Brillouin and XRD measurements, to the best of our knowledge, there is no standard laboratory-based Brillouin setup inherently equipped with an integrated XRD system.

Given that synchrotron XRD is not readily available for routine use and that Brillouin measurements require extensive acquisition times - often ranging from one to three days per single measurements at one out of several angular orientation - conducting XRD at every pressure step would significantly extend the duration of the study, making it impractical within the timeframe of a PhD project. While XRD-based pressure determination is indeed a robust method, it was not feasible at every stage of our experiment. Considering that Raman edge calibration remains a widely accepted approach in high-pressure research, we are confident that our pressure determinations based on the Raman T_{2g} mode provide reliable results.

Unlike single-crystal studies, which reflect directional properties, our polycrystalline measurements average these effects. Under compression, single crystals may fracture or transform into polycrystalline forms. For example, stishovite crystals are believed to transform into polycrystalline aggregates at depths greater than 1500 km due to the spontaneous distortion of the crystals once a pressure threshold is surpassed, triggering a phase transition (e.g. Buchen et al., 2018). A similar behavior has been observed in single crystals of argon, which transformed into polycrystalline aggregates at 4.3 GPa (Chen et al., 2010; Grimsditch, 1986; Marquardt et al., 2013; Shimizu et al., 2001).

Thirdly, literature data for stishovite and post-stishovite phases are not compared properly. There are very high-quality data and even single crystal C_{ij} data in the literature that are not even cited and discussed in Figure 2 and relevant sections. It's not clear why the authors did not even do so but this reads like a very bad practice in science.

Indeed, the data from (e.g. Zhang et al., 2021) are now included in our revised Fig. 2. We aimed to compare conditions similar to ours, which are based on polycrystalline samples, and therefore omitted comparisons with single-crystal data. This revision has been incorporated into the updated manuscript.

Fourthly, silica phases always contain some Al so Al effects on elasticity should be considered. In fact, there are also rich literature on this issue that was totally ignored in this paper. Lastly, the elasticity and thermodynamic modelling should incorporate full datasets including V_s , V_p , and density. There're no temperature effects measured and addressed here (no new data to address this issue). Based on what we have in the literature, HT effect at HP appears to be a major uncertainty, rather than the data the authors reported. Given all these considerations, there are just huge error bars that would mask out all effects and applications that the authors are trying to address. Figure 3 does not even show uncertainties?

At the end, the paper has a nice story to tell but without much of scientific support.

Given the extensive time and effort dedicated to this study on pure SiO_2 samples, incorporating additional data, such as Al- SiO_2 , probably with varying concentrations, could indeed provide further insights but would extend beyond the intended scope of this work. As noted by Reviewer #2, the limited data available for SiO_2 at ultrahigh pressures reflects the significant challenges and time

investment required for such measurements. Our primary objective was to first establish a robust understanding of the pure SiO₂ system, providing a foundation for future studies that may explore related compositions in greater detail.

Our modeling aimed to examine the effect of two assumed temperature profiles, potentially representing lower mantle conditions, e.g. cold slab- (Lin et al., 2022) and lower mantle geotherm (Katsura 2022), on the shear wave velocity (V_S) of MORB. The revised manuscript also includes the effect of ambient temperature. Density and V_P were not the focus of this study, as knowledge of V_S alone provides crucial insights into the geophysical implications of reduced shear wave velocities observed in regions of LLSVPs. The magnitude of these reductions spans from $-(0.5-1)\%$ (Cottaar et al., 2016, Jones et al., 2020, Lay et al., 2006, Ohta et al., 2008, Lekic et al., 2012) in the shallowest depths of LLSVPs, within 2450-2520 km (Lay et al., 2006, Ohta et al., 2008) to $-(1.5-2)\%$ (Konishi et al., 2009, Kawai et al., 2010, Ballmer et al., 2016, Lekic et al., 2012), within 2500-2600 km (Konishi et al., 2009, Ballmer et al., 2016), or 2471-2547 km (Kawai et al., 2010), reaching up to -3% at the very bottom of the lower mantle (Fan et al., 2021, Shephard et al., 2017, Wen et al., 2002), within 2520-2620 km (Fan et al., 2021), ~ 2550 km (Shephard et al., 2017) or ~ 2600 km (Wen et al., 2002). Thus, understanding the V_S of the SiO₂ phase is key to assessing its effect and contribution in a MORB composition in the deep mantle.

This study provides key experimentally-determined elastic parameters for SiO₂ phases, including the shear modulus modulus (G_0) and its pressure derivative (G'), which are critical for understanding SiO₂'s geophysical implications in the lower mantle.

However, due to the challenges of high-temperature experiments, it is common practice to use thermodynamic parameters from the self-consistent framework developed by (Stixrude et al., 2011). This approach is widely adopted in mineral physics to interpret the effects of elasticity parameters on the V_S profile of MORB under high-pressure and high-temperature conditions.

While (Mattern et al., 2005) emphasized that pressure effects dominate over temperature corrections in influencing V_S , the role of G' remains critical for interpreting seismic observations. Previous studies (Murakami et al., 2012, Stixrude et al., 2011) highlight that temperature sensitivity is pronounced in parameters like η_{50} , particularly for bridgmanite and MgO, with experimental uncertainties around 10%. In our revised manuscript, we consider this variation, which corresponds to a ± 1 vol.% variation in MORB content needed to explain observed $-(1.5-3)\%$ seismic anomalies.

The lack of experimentally determined high-temperature elasticity data remains challenging in mineral physics. Common practice adopted by different researchers include calculation/modeling of this data from the available theoretical study of (Stixrude et al., 2005, 2011, 2013). This approach was adopted for example by (Wang et al. 2020, Nature Communications), who assessed the effect of the CF phase, purely theoretically, in a MORB composition, by adopting assumed temperature profiles along the lower mantle. Another study that modeled the effect of temperature on the V_S profile in CaPv and MORB is (Thomson et al., 2019, Nature). Similarly, such calculations/modeling based on theoretical framework from (Stixrude et al. 2011) were performed in our work under assumed ambient temperature, cold slab and lower mantle conditions. Therefore, we followed the approach from recent studies.

Fig. 3, currently Fig. 4, includes uncertainties in our modeled V_S profiles for the cold slab- and lower mantle geotherms, with error bars reflecting potential variations, both in SiO₂ system and MORB assemblage. As mentioned above, the greatest temperature sensitivity of $\pm 10\%$ in η_{50} , would result in

a ± 1 vol.% variation in our modeled MORB content needed to explain the observed $-(1.5-3)\%$ discontinuities.

In recent years, multiple studies have proposed different explanations for the negative shear wave anomalies within LLSVPs. Our manuscript discusses these perspectives while considering the potential contributions of SiO_2 phases. We believe that our study, which presents the first experimentally determined shear modulus (G) and its pressure derivative (G') in high pressure SiO_2 phases, makes a significant contribution to understanding LLSVPs. These novel results provide new insights into an ongoing geophysical debate.

Please find below our responses. Questions/Comments from the reviewers are in black, our replies are in blue, and modifications to the manuscript in green. All the lines indicated below are in the revised manuscript with changes incorporated.

Reviewer #1

Manuscript COMMSENV-24-3193A (Revised manuscript)

‘Ultrahigh-Pressure Sound Velocities of Dense SiO₂ Phases and the Origin of the LLSVPs’

by Krymarys et al.

While the authors have made efforts to address most of the points raised during the review, many of the points have not been adequately handled and some inconsistencies still persist. Unfortunately, some of the responses are unnecessarily lengthy and do not contribute to clarify the points because the changes in the actual manuscript text are sometimes minimal. Also, some of the additions in the text in response to the comments are misplaced and seriously affect the flow of the manuscript. Therefore, I cannot recommend the publication of the manuscript in the current state.

We would like to thank Reviewer #1 for providing us with feedback on our manuscript and for the opportunity to revise our manuscript. We recognize that some aspects of our previous response may not have fully addressed the reviewer’s concerns. In light of the renewed comments, we have revised our responses to provide clearer and more concise explanations. We have also refined the manuscript accordingly to ensure that each point is addressed directly and transparently. Below, we provide point-by-point responses to the comments and suggestions raised by Reviewer #1.

Here are some of the points of concern that have not sufficiently or adequately clarified:

1) Lines 155-168: The discussion about the interferometer models in the comparison of the present results with those from Asahara’s et al (2013) data is very technical and unnecessary as it does not bring much to explain the divergence as the velocities should remain the same regardless of the instrument generation (as acknowledged by the authors...). Therefore, the addition is unnecessary and lines 155 to 164 should be removed to focus the discussion on more realistic explanations for the disagreement.

Following this suggestion from Reviewer #1, the information related to the interferometer in our revised manuscript has been removed. We discuss instead about the lack of angular variation in measurements as the primary reason for the velocity discrepancy between the results of Asahara et al., 2013 and our own.

As briefly stated in our previous rebuttal, Asahara et al., 2013 performed only a single velocity measurement at each pressure point, keeping the cell orientation fixed throughout. In contrast, we conducted multiple velocity measurements at each pressure point using different cell orientations.

When velocity measurements are conducted at a fixed angle throughout a single series of measurements, the fundamental assumption in polycrystalline measurements - that the sample is a randomly oriented, fine-grained aggregate - may no longer hold. This is especially true if the grain size or crystallographic texture has developed significantly within the sample. In such cases, there is a considerable risk that the measured velocities will deviate substantially

from the true average velocity representative of a polycrystalline aggregate. Indeed, this likely explains the significantly elevated velocities reported by Asahara et al., 2013.

The reason Asahara et al., 2013 limited their measurements to a single fixed orientation was due to practical constraints: they were using a Brillouin scattering system installed inside a synchrotron beamline hatch, where limited beamtime necessitated a more time-efficient measurement protocol. However, this approach is not consistent with the goal of obtaining the most reliable and representative velocity data.

In contrast, to minimize such concerns, we performed measurements at multiple orientations under the same pressure condition. This practice has been routinely adopted in previous studies using Brillouin scattering on polycrystalline samples, and we believe this approach improves the reliability of our measurements compared to those of Asahara et al., 2013, whose methodology did not include this standard step.

We should also notice that due to differences in synthesis conditions, pressure-loading protocols, laser annealing, and other experimental details in Asahara et al., 2013 DAC experiments, the grain size, crystallographic preferred orientation, and microstructure of their sample are likely to differ from ours. Therefore, a direct comparison of these specific aspects between their sample and ours is not straightforward.

A revised discussion can therefore be found in the updated lines 172-185 of our manuscript, which now include the following text: “Asahara et al., 2013²⁸ performed only a single velocity measurement at each pressure point, keeping the cell orientation fixed throughout. In contrast, we conducted multiple velocity measurements at each pressure point using different cell orientations. When velocity measurements are conducted at a fixed angle throughout a single series of measurements, the fundamental assumption in polycrystalline measurements - that the sample is a randomly oriented, fine-grained aggregate - may no longer hold. This is especially true if the grain size or crystallographic texture has developed significantly within the sample. In such cases, there is a considerable risk that the measured velocities will deviate substantially from the true average velocity representative of a polycrystalline aggregate. In contrast, to minimize such concerns, we performed measurements at multiple orientations under the same pressure condition. This practice has been routinely adopted in previous studies using Brillouin scattering on polycrystalline samples, and we believe this approach improves the reliability of our measurements compared to those of Asahara et al., 2013²⁸, whose methodology did not include this standard step”.

2) Following up the previous point, the sentence in lines 164- (‘Therefore, the results of Asahara et al. 2013 may not fully account for the effects of elastic anisotropy, potentially leading to an overestimated dataset compared to our result’) is unclear and should be reworked. In fact, I found much more convincing the arguments in the response that the additions in the manuscript related to this point. Among the possible arguments discussed, only potentially the limited sampling in Asahara et al (2013) is a solid argument but it is difficult to read this from the above-mentioned sentence in the text... Also, I still think that the authors should be more conservative in the estimation of errors to account from all plausible sources of uncertainty as a difference of 150 m/s is not unrealistic when dealing with non-annealed polycrystalline materials at the extreme high pressures of this study.

First of all, we sincerely thank Reviewer #1 for the thoughtful and constructive comments on this issue. We are also pleased that Reviewer #1 recognized the robustness of our approach - particularly our use of multiple measurement orientations - as an important distinction from the work of Asahara et al., 2013. As noted by the Reviewer #1, we acknowledge that our previous response and the manuscript did not sufficiently highlight or directly address these points, and we have revised both accordingly.

Our response to this Reviewer #1's concern is presented in the newly revised manuscript immediately after our response to the previous concern, at lines 185-189, where the indicated sentence has been reworked to the following: "Therefore, the results of Asahara et al., 2013²⁸ may have been influenced by the development of a preferred crystallographic orientation or texturing, potentially explaining the observed ~0.3 km/s higher average V_S profile over ~30-130 GPa (Fig. 2). The differences are less discernible at lower pressures but become more pronounced in the extended range of ~60-130 GPa".

With regard to error estimation, we believe that we had already attempted to incorporate what we considered the primary source of uncertainty - namely, the variation in measured velocities due to differences in measurement orientation. However, following the reviewer's suggestion, we have further examined additional plausible and quantifiable sources of error.

Although it is not straightforward to directly quantify the effects of non-annealing as a velocity uncertainty, we considered that stress - specifically, deviatoric stress - might serve as a measurable proxy. Therefore, we evaluated how the uncertainty in pressure associated with deviatoric stress (5 GPa deviatoric stress for CaCl₂-type at 92 GPa and 6 GPa deviatoric stress for α -PbO₂-type at 99 GPa - Fig. S1-S2, Supplementary Text 5) could translate into uncertainty in velocity. Based on this analysis, the resulting maximum velocity error was estimated to be ± 0.05 km/s.

To be even more conservative in our error estimation, we also investigated the impact of background signal subtraction during spectral analysis as another quantifiable source of error. This yielded a maximum uncertainty of ~0.2 km/s.

These additional sources of error are relatively minor compared to the angular uncertainties discussed and, in our view, are unlikely to fully explain the discrepancy between our results and those of Asahara et al., 2013.

3) While I appreciate the efforts to clarify the temperature projection of the velocities in MORB, the text in lines 288-318 is clearly misplaced in the manuscript and cuts the flow of the discussion. It should not be included in the discussion about the origin of the LLSVP but rather integrated in the section related to modelling the V_S profiles of MORB.

We are very grateful to the Reviewer #1 for carefully examining the logical structure of our manuscript.

In response to this comment, the paragraph has been moved to the newly added section, "Comparison with theoretical models and experimental data," (line 255) positioned directly after the two-preceding modeling-related sections, namely "Modeling of V_S profile of MORB" and "Comparison with PREM." This restructuring improves the overall flow and coherence of the revised manuscript. By adding this section, we provide a more structured progression -

first outlining the modeling procedure, then discussing its relationship to PREM, and finally comparing our results with previous data (theoretical and experimental), along with further considerations such as variations in mineral phase proportions.

We strongly believe that the changes we have made now allow readers to follow how the effects of temperature were incorporated, without disrupting the overall logical flow of the paper.

4) Line 144. – should be ‘X-ray diffraction’. However, the reference from Shieh et al 2022 is not appropriate here because the direct data from their XRD study does not show any evidence for softening (i.e., average $Q(hkl)$ increases with increasing pressure, even across the ferroelastic transition), it is only when the differential stress t is calculated, which involves G , that there is softening. Therefore, the softening is related to G (which is not obtained in their study) not to the actual parameters measured in the study and hence, the XRD study should not be mentioned here as evidence for softening in experimental studies. In my opinion the paper by Zhang et al. (2021) is already providing the clearest experimental evidence for softening.

In response to this comment, we have removed the reference to Shieh et al. 2002 from the revised manuscript. We agree that Zhang et al. 2021 provides more direct and relevant experimental evidence for softening, and we have retained and emphasized this reference to better support our discussion. Our revisions are present at lines 160-161. “This is anticipated from theory to induce a sharp shear softening²⁴, and was also shown in an X-ray diffraction and Brillouin study of a single-crystal stishovite at 55 GPa by Zhang et al., 2021⁴⁴ or polycrystalline stishovite from Brillouin study of Asahara et al., 2013²⁸ at the pressure range of 25-35 GPa”.

There are other additions/responses related to the comments by Reviewer 2 (that I have been asked to check) that are only partially addressed.

1) The lengthy response related to the effects of stress and texture in the Brillouin signal does not unfortunately help to clarify this point to potential readers because these arguments are not included in the text (only partially in the Supplementary Materials). I would recommend that authors directly address these concerns in the main text (in a more direct and concise manner), possibly in the section explaining the measurements, as a disclaimer to remove all potential concerns for the rest of the discussion.

Following Reviewer #1’s suggestion, the revised manuscript includes information related to the effects of stress and texture in the Brillouin signal.

The part describing the Brillouin measurements has been concisely elaborated and it is present in the revised lines 125-135 as follows:

“Fig. 1 shows the representative high-pressure raw Brillouin scattering spectra from the two synthesized high-pressure SiO_2 phases. These spectra were acquired in two independent series of measurements (Table S1, Fig. S6), using centrally positioned samples in the DAC chambers. Although the samples were not annealed, stress conditions were evaluated at the synchrotron

for the CaCl₂-type and α-PbO₂-type phases at 92 and 99 GPa, respectively, indicating deviatoric stresses of ~5-6 GPa (Fig. S1-S2). To ensure the reliability of Brillouin data under ultrahigh-pressure conditions, we collected sharp, high-quality peaks by measuring multiple angular orientations and extending acquisition times over several days (Fig. 1, Table S1). Since deviatoric stress typically induces peak broadening in Brillouin spectra, the absence of systematic broadening in our data suggests that stress did not progressively increase with pressure. Both the lowest- and highest- pressure data points in Fig. 1 are within ±0.1 of the average FWHM value, consistent with the high-pressure synchrotron data indicating ~5-6 GPa of stress (Fig. 1, Fig. S1-S2, Supplementary Text 5)”.

Additionally, lines 109-111 have been revised to clarify that synchrotron X-ray diffraction (XRD) was used to determine both structure and stress conditions of our samples under high-pressure conditions.

2) Concerning the pressure determination, I agree with the authors that access to XRD analysis for pressure calibration are not routinely available but they should not claim that there is no standard laboratory-based Brillouin setup inherently equipped with an integrated XRD system, because a system of this type is operated for more than 10 years at BGI-University of Bayreuth. Nevertheless, I would recommend that the authors acknowledge the limitations of the Raman edge approach and the large uncertainties associated to the method particularly in the pressure range below Mbar. Did the authors cross-checked the pressure given by the diamond edge and calculated from the EoS of B2-NaCl in the few XRD data that the authors could collect? This would be a good support to demonstrate that the pressure is properly determined (or at least to have an idea of the deviation).

Following the reviewer comment we confirm that BGI-University of Bayreuth is one of the leading institutions in the field of mineral physics, with a well-established and continuously developing infrastructure. We apologize for the previous notations regarding the existence of such a setup.

Regarding the use of the Raman edge approach, we provide more clarification. Although the Raman T_{2g} mode is well calibrated up to 410 GPa (Akahama and Kawamura, 2010), its accuracy at the megabar scale can be affected by stress and reduced edge sharpness, particularly above ~200 GPa (Akahama and Kawamura, 2006, 2007, 2010). In this study, all T_{2g} peaks remained sharp, enabling reliable pressure determination across the investigated range of ~40-150 GPa. To further validate the pressure estimates, we cross-checked values using the equation of state (EoS) of the NaCl B2 phase via synchrotron XRD. Comparison revealed that the Raman T_{2g} method yielded pressures up to 2 GPa lower than those obtained from XRD for both CaCl₂-type and α-PbO₂-type SiO₂ phases.

Following Reviewer #1's suggestion, we have addressed the limitations of the Raman T_{2g} mode in the revised manuscript and clarified its potential deviation from XRD-based pressure determination. These revisions are present at lines 135-142 of our newly revised manuscript as follows:

“For each Brillouin pressure point (Table S1), pressure was determined using the Raman T_{2g} mode, measured at several spots within the central ~20 μm region of the probed sample both before and after each Brillouin acquisition. The values were averaged, and the standard

deviation is reported in Table S1. A cross-check between Raman T_{2g} -derived pressures and those obtained from the equation of state (EoS) of the NaCl B2 phase using synchrotron XRD at 92 GPa and 99 GPa (for the CaCl_2 -type and α - PbO_2 -type SiO_2 phases, respectively) showed that Raman-based pressures were up to 2 GPa lower at these high-pressure points”.

3) Regarding the role of Al, without asking for the experimental results, I think the author could include a comment on this in the discussion about the origin of the LLSVPs, as there is enough information about the effect of Al (changing the transition pressures, decreasing the shear modulus) for a qualitative discussion.

In response to this suggestion from Reviewer #1, we have revised the final paragraph of the manuscript to expand the discussion on the potential effects of aluminum, particularly its role in shifting transition pressures and reducing elastic moduli.

In more realistic MORB compositions, Al incorporation into stishovite can reach ~2-2.5 wt.% at 2000 °C (Liu et al., 2006), with similar levels reported in other studies (Irifune & Ringwood, 1993; Jiang et al., 2009; Lakshtanov et al., 2007; Litasov et al., 2007; Liu et al., 2006, 2007; Ono, 1998, 1999; Pawley et al., 1993). Higher Al_2O_3 contents have been observed under extreme conditions - up to 3.4 wt.% in CaCl_2 -type SiO_2 at 60 GPa and as much as 12.6 wt.% in α - PbO_2 -type SiO_2 at 113 GPa (Hirose et al., 2005).

The presence of Al_2O_3 has been shown to lower the stishovite – CaCl_2 -type transition pressure (e.g., Lakshtanov et al., 2007a; Zhang et al., 2022; Criniti et al., 2023; Bolfan-Casanova et al., 2009) and reduce the bulk modulus (K) of Al-bearing phases (e.g., Andraut et al., 2003, 2014; Grocholski et al., 2013; Ono et al., 2002; Panero et al., 2003). However, experimental constraints on its effect on the shear modulus (G) remain limited, with the only available data coming from a narrow 10 GPa pressure window for Al-bearing CaCl_2 -type SiO_2 containing water (Lakshtanov et al., 2007b).

The paragraph concisely captures these aspects. These revisions are present at lines 379-387 of our newly revised manuscript as follows: “The long-term accumulation of the chemically distinct subducted oceanic crust in the lowermost mantle is thus anticipated to alter the bulk chemistry of the lower mantle towards more SiO_2 - and Al_2O_3 -rich composition over the subduction history. The effect of Al_2O_3 across stishovite to CaCl_2 -type SiO_2 phase transition was shown to reduce the transition pressure (e.g. ⁸⁰⁻⁸³) and to decrease the bulk modulus (K) of SiO_2 phases (e.g. ^{48,84-87}). However, experimental constraints on its effect on the shear modulus (G) remain limited. The only available data, from a narrow ~10 GPa pressure range for Al-bearing CaCl_2 -type SiO_2 containing water⁸⁵, suggest a decrease in G . Nevertheless, the exact role of Al_2O_3 on the V_s profiles of CaCl_2 -type and α - PbO_2 -type SiO_2 remains unknown and would require future experimental studies to quantify this effect”.

In summary, the authors should still rework the main text to clarify the most important points that seems to be better handled in the responses than in the text. This approach does not contribute to the clarity of the manuscript.

We thank Reviewer #1 for the time and effort dedicated to reviewing our work and for the opportunity to further improve the quality and clarity of our manuscript.

Please find below our responses. Questions/Comments from the reviewers are in **black**, our reply is in **blue**, and modifications to the manuscript in **green**. All the lines indicated below are in the revised manuscript with changes incorporated.

Reviewer #3 (Remarks to the Author):

After carefully evaluating the authors' rebuttal and the revised manuscript, I find that my principal concern has not been satisfactorily addressed. Consequently, I am unable to recommend publication in its present form until this issue is fully resolved. I therefore restate my primary concern below to ensure complete clarity.

We would like to thank Reviewer #3 for providing us with feedback and for giving us an opportunity to revise our work. We also apologize if our previous rebuttal did not fully address the Reviewer #3's concerns. In this revision, we have made every effort to respond to the Reviewer #3's comments in a more direct and concise manner.

We believe that these improvements have resulted in a clearer and more coherent logical structure, which will allow readers to follow the manuscript more smoothly. Below, we provide point-by-point responses to the aspects raised by Reviewer #3.

In the revised manuscript, the authors continue to use $\delta \ln V_s$ to denote both the lateral shear wave velocity anomalies of LLSVPs and the velocity decrease produced by the CaCl_2 type $\text{SiO}_2 \rightarrow$ seifertite phase transition, which makes the interpretation confusing. They are different concepts and are used inappropriately. For example, in lines 229–232 the authors state, "It is generally accepted from previous shear wave tomographic observations that the negative shear velocity anomalies in LLSVPs exhibit depth-dependent variations with anomalies ranging from approximately -0.5% to -1% in shallow regions to around -3% at depths ranging from 100 to 200 km from the CMB." These values represent velocity anomalies of LLSVPs relative to reference models such as PREM. The authors use the term "anomaly" in line 235 to describe the -1.5 % anomaly correctly; however, in lines 239, 247, 257, and 267 they use "discontinuity" to refer to the same velocity anomalies. These inconsistencies suggest that the authors remain unclear about the distinction between velocity anomalies ($\delta \ln V_s$) and the velocity contrast across a discontinuity (for which I recommend an alternative notation, such as ΔV_s , to distinguish two concepts).

We apologize for not fully addressing the Reviewer #3's concerns in the previous round of revision. In this revised version, we have carefully followed the Reviewer #3's suggestion: the term "**seismic anomaly**" (e.g., $\delta \ln V_s$) is now exclusively used to describe seismologically observed anomalies, while the term "**discontinuity**" (e.g., ΔV_s) is reserved for velocity discontinuities associated with discontinuous feature of MORB where SiO_2 phase transition occurs.

We have consistently maintained this distinction throughout the Manuscript and Supplement. Specifically, $\delta \ln V_s$ is now used to refer to seismically observed velocity

anomalies relative to PREM, and ΔV_S to describe the modeled contrast (across the discontinuity), where SiO_2 phase transition takes place.

These changes are reflected in the revised lines:

- Manuscript: 152, 154, 200 (“anomalies” replaced with “profiles”), 206-207, 219, 221-224, 230, 240, 243, 260, 271-272, 278-280, 282-287, 291, 293, 298-301, 304, 317, 325, 335, 345, 665-666
- Supplement: 283, 308-309, 312, 316-317, 319, 322, 323-324, 326, 328, 338-340, Table S9

On the other hand, the ΔV_S generated by the phase transition does not play a significant role in explaining the velocity anomaly ($\delta \ln V_S$) in LLSVP based on their results, because its effect in MORB is only $\sim 0.6\%$ without SiO_2 enrichment (Table S6-S7), and its contribution is further decreased when considering the MORB fraction. To illustrate the limited contribution, I recommend adding a representative V_S profile of the assemblage containing 23 vol % MORB (choose one case from Table S9) to Figure 4; In this case, the expected differences caused by the phase transition should be negligible.

We appreciate the Reviewer #3’s insightful comment regarding the limited contribution of the SiO_2 phase transition to the overall shear wave velocity anomaly ($\delta \ln V_S$) in LLSVP.

We agree that the ΔV_S generated by the CaCl_2 -type to $\alpha\text{-PbO}_2$ -type transition in MORB is modest $\sim 0.6\%$ without SiO_2 enrichment, as shown in Tables S6-S7, and that this contribution becomes even smaller when MORB is SiO_2 depleted. Bridgmanite proportion enrichment would likely reduce the influence of this transition even further.

To illustrate this limited effect, our manuscript presents Fig. 4 with a representative, however modest, case of 33 vol.% MORB along the cold slab geotherm and 28 vol.% MORB along the lower mantle geotherm, using the MORB composition from Table S3 at 60 GPa. These profiles - without SiO_2 enrichment - also incorporate cubic Ca-perovskite data from (Greux et al. 2019), and result in ΔV_S values of $\sim 0.6\%$, aligning with the Reviewer #3’s interpretation. Additionally, Figure 3 shows that the SiO_2 phase transition in MORB can generate ΔV_S of only $\sim 0.33\%$ at ambient temperature. To make these subtle differences clearer to the reader, we have zoomed in on the relevant regions of Figures 3B and 4B, which visually highlight the close proximity of the V_S profiles across the CaCl_2 - to $\alpha\text{-PbO}_2$ -type transition.

To better highlight the limited effect of SiO_2 phase transition itself, we have provided a clarification in our manuscript, revised lines 222-229.

“Our experimental results reveal that the CaCl_2 - $\alpha\text{-PbO}_2$ phase transition in SiO_2 produces a negative velocity discontinuity. At the transition pressure in pure SiO_2 , along the cold slab geotherm, this discontinuity corresponds to a decrease of approximately 3% in shear wave velocity. However, when considered in the context of MORB compositions, this reduction is mitigated to about 0.6%, which is relatively

minor compared to the overall discrepancy of ~7-14% between the experimentally determined SiO₂ velocity profile and theoretical predictions. This highlights the significant role of the overall velocity reduction in SiO₂, which has a greater impact on the seismic velocity profile than the phase transition alone”.

However, this does not undermine the importance of this work to a large extent. The main contribution of this work is that the measured V_s is significantly lower than previous estimates, although the high-temperature extrapolation introduces some uncertainty. Therefore, the authors should focus on the velocity profiles with varying MORB fractions relative to PREM, i.e. results in Table S9, which is more important to explain the depth-dependent velocity anomalies from -1.0% to -3.0%. According to Table S9, the change in velocity anomalies in LLSVP primarily results from the effect of temperature and MORB fraction, rather than the phase transition. And the estimated required MORB fractions based on their data should be less than those based on previous data.

We thank the Reviewer #3 for feedback and suggestions. We agree that the main strength of this work lies in our experimentally constrained shear wave velocity (V_s), which is significantly lower than previous estimates, although the high-temperature extrapolation can introduce some uncertainty, as discussed in our manuscript.

To address the Reviewer #3's comment, we emphasize that Table S9 indeed is an important part of our analysis, highlighting how temperature and MORB fraction - rather than the SiO₂ phase transition - can explain the observed velocity anomalies ($\delta \ln V_s$) in LLSVPs.

Our results show that increasing temperature (≥ 3000 K) enhances the predicted ΔV_s from approximately -0.6% along cold slab geotherm to:

~-1% when using experimental bridgmanite endmembers without SiO₂ enrichment,

~-3% when SiO₂ enrichment is included.

As correctly noted by the Reviewer #3, the direct effect of the generated ΔV_s , where SiO₂ phase transition occurs, as the negative shear velocity contrast across a discontinuity feature in MORB, remains modest ~-0.33% at ambient temperature and ~-0.6% under cold slab- or lower mantle geotherms - when no SiO₂ enrichment is assumed.

The revised manuscript, in addition to the discussion on CaPv and SiO₂ phases, discusses the implications of increasing temperature on the reduction of vol.% of MORB required to explain the observed negative anomalies ($\delta \ln V_s$) of -(1.5-3)%. Revised lines 282-295.

"Our study presents a comprehensive analysis of the potential discontinuous feature associated with MORB in the lower mantle (Table S9). The MORB volume fractions required to explain the observed seismic velocity anomalies are significantly lower than those proposed by previous studies - for example, Thomson et al., 2019²⁷ estimated up to 64% MORB at 100 GPa and 48% at 125 GPa to explain a $\delta \ln V_s$ of -1.5%. In contrast, our results indicate that 23-33 vol.% MORB may already account for a $\delta \ln V_s$ of -1.5% (Table S9, Fig. 4). A more pronounced anomaly $\delta \ln V_s$ of -3%

would require a higher fraction of ~47-66 vol.%. Importantly, these values decrease under elevated temperatures exceeding 3000 K (Table S9). For instance, if lower mantle temperatures range from 2600 K at the top of the D" layer to 4000 K at the core-mantle boundary, as suggested by Manthilake et al., 2011⁶³, the modeled shear wave velocity contrast (ΔV_S) could increase even by a factor of two (e.g., from ~-0.6% at <2600 K to ~-0.7% at 3000 K and ~-1% at 4000 K). This increase in ΔV_S would in turn reduce the required MORB volume fraction - for example, from 33 vol.% under a cold slab geotherm (Case 2A) to 22 vol.% at 3000 K and as low as 14 vol.% at 4000 K (Table S9)".

Additionally, the role of the SiO₂ phase transition in contributing to the discontinuous feature in MORB has been more carefully contextualized in our revised manuscript. Revised lines 231, 263-264, 279, 298-300.

Issues with the citations still exist. For example, as I noted in my previous review, Yang et al. (2014) does not contain information on the CaCl₂ type SiO₂ → seifertite transition, whereas Karki et al. (2001) does, but the revised manuscript cites Yang et al. (2014) rather than Karki et al. (2001) at line 99. The authors need to examine every reference carefully to ensure each citation is accurate and appropriate.

We apologize for this confusion. In fact, Karki et al., 2001 as intended but awkwardly misplaced in the previous version, is the pioneering theoretical study predicting the V_S of mineral phases, including SiO₂. Therefore, the revised version includes only Karki et al., 2001 citation as a proper attribution. Revised lines 99, 160.

We thank Reviewer #3 for the time and effort dedicated to improving our work. We believe that the revised version of the manuscript has benefited significantly from the provided insightful comments and suggestions.

Reviewer #1 (Remarks to the Author):

Overall, I am satisfied with the corrections implemented in the manuscript.

There is a last imprecision that should be corrected in the discussion of the effect of Al₂O₃ on the elasticity of SiO₂ stishovite. Ref. 85 is inappropriate here because it is a room pressure study of the effect of Al³⁺ on the elasticity of SiO₂-stishovite. It is not clear what is the 10 GPa range the authors refer to and therefore the statement and the discussion should be modified for correctness.

We thank Reviewer #1 for this important observation and apologize for the imprecision. Indeed, the study by Lakshtanov et al., 2007 published in *American Mineralogist* focuses exclusively on the elastic properties of Al- and H-bearing stishovite at room pressure up to ~25 GPa, and therefore is not appropriate for discussing the shear modulus of the CaCl₂-type phase. We acknowledge that the reference was incorrectly cited in this context.

In fact, Lakshtanov et al., 2007 also published a separate study in *PNAS*, which extends up to 45 GPa and investigates the stishovite to CaCl₂-type phase transition, but does not provide the shear modulus (*G*) for Al-bearing CaCl₂-type SiO₂. The shear properties of Al-bearing CaCl₂-type or α-PbO₂-type SiO₂ phases remain experimentally unconstrained. For example, Lakshtanov et al., 2007 (*Am. Mineral.*) investigated Al- and H-bearing stishovite at room pressure up to ~25 GPa and reported a decrease in shear modulus (*G*) compared to pure room pressure SiO₂ (e.g., Jiang et al., 2009). These results, however, are limited to the stishovite stability field and involve compositions containing minor water. The current absence of experimental data on the shear modulus (*G*) and its pressure derivative (*G'*) for Al-bearing CaCl₂-type and α-PbO₂-type SiO₂ phases, particularly under anhydrous conditions, highlights the importance of further high-pressure elasticity studies to quantify the effect of Al₂O₃ on the seismic properties of these high-pressure SiO₂ phases.

The revised manuscript incorporates these changes at lines 380-389: "However, the shear properties of Al-bearing CaCl₂-type or α-PbO₂-type SiO₂ phases remain experimentally unconstrained. For example, Lakshtanov et al., 2007⁸⁵ investigated Al- and H-bearing stishovite at room pressure up to ~25 GPa and reported a decrease in shear modulus (*G*) compared to pure room pressure SiO₂ (e.g., Jiang et al., 2009⁸⁸). These results, however, are limited to the stishovite stability field and involve compositions containing minor water. The current absence of experimental data on the shear modulus (*G*) and its pressure derivative (*G'*) for Al-bearing CaCl₂-type and α-PbO₂-type SiO₂ phases, particularly under anhydrous conditions, highlights the importance of further high-pressure elasticity studies to quantify the effect of Al₂O₃ on the seismic properties of these high-pressure SiO₂ phases".